



# On the role of the seawater absorption to attenuation ratio in the radiance polarization above the Southern Baltic surface

Włodzimierz Freda[1], Kamila Haule[1], Sławomir Sagan[2]

[1]Department of Physics, Gdynia Maritime University, Gdynia, 81-225, Poland
[2]Institute of Oceanology, Polish Academy of Sciences, Sopot, 81-712, Poland

*Correspondence to*: Włodzimierz Freda (wfreda@am.gdynia.pl)

**Abstract.** Information about polarization of light leaving the ocean surface has the potential to improve the quality of the bio-optical parameters retrieval from ocean color remote sensing (OCRS). This improvement can be applied in numerous ways such as limiting of sun glints and obtaining information about atmospheric aerosol properties for atmospheric correction as well as increasing the accuracy of the algorithms based on the water-leaving signal. Polarization signals at the top of the atmosphere (ToA) that include the water-leaving signal are strongly influenced by atmospheric molecular scattering and by direct sun and sky reflections from the sea surface. For these reasons, it is necessary to better understand the factors that change the polarization of light in the atmosphere-ocean system, especially in coastal zones affected by dynamic changes. In this paper, the influence of seasonal variability of light absorption and scattering coefficients (inherent optical properties, IOPs) of seawater, wind speed and solar zenith angle (SZA) on the polarization of upwelling radiance over the sea surface in the visible light bands is discussed. The results come from a polarized radiative transfer model based on the Monte Carlo code and applied to the atmosphere-ocean system using averaged IOPs as input data. The results, presented in the form of polar plots of the total upwelling radiance degree of polarization (DoP), indicate that regardless of the wavelength of light and type of water, the highest value of the above water DoP is strongly correlated with the absorption-to-attenuation ratio. The correlation is a power function that depends on both the SZA and the wind speed. The correlation versatility for different wavelengths of light is very unusual in optics of the sea and is therefore worth emphasizing.

## 1 Introduction

Satellite ocean color radiometry has been developed for decades to study the interaction of a light field within the visible part of the spectrum (i.e., 400-700 nm) with the different optically significant constituents of sea water. The research has been focusing on information coming from the intensity of water-leaving light - its measurement, retrieval, correlations and interpretation (e.g., Volpe et al., 2012; Zibordi et al., 2013; Sammartino et al., 2015). However, in addition to the light intensity, consideration of light polarization has been demonstrated to improve the accuracy of the information from a variety of remote sensing applications, i.e., in remote radar measurements (Hajnsek et al., 2003; Soloviev et al., 2012;



Benassai et al., 2013), in atmospheric correction algorithms (Chowdhary et al., 2002), in above-surface measurements (Garaba and Zielinski, 2013) and in beneath-surface measurements (Ibrahim et al., 2012).

Theoretical and field measurements have shown that the polarization of the underwater light field is sensitive to the nature of the suspended marine particles. Ibrahim et al. (2012) demonstrated that the attenuation-to-absorption ratio influences the polarization of upwelling radiance below the sea surface. Polarized measurements have also been performed near and above the sea surface. Reduction of sun glints to improve the ocean color retrieval has been studied by Zhou et al. (2017) as well as Garaba and Zielinski (2013) and limitation of sky reflections by observation of sea surface at the Brewster angle from the ship board has been examined by Wood and Cunningham (2001) and by Cunningham et al. (2002). Polarization distribution of skylight reflected off the rough sea surface has been examined by Zhou et al. (2013). They simulated the degree of polarization as well as the angle of polarization (AoP) for reflected parts of upwelling radiance and discussed its variability with SZA for 0°, 30°, 60° and 90°. Moreover, Zhou et al. (2013) showed the influence of wind speed and direction on the polarization pattern of reflected radiance. Radiative transfer simulations have shown the effect of marine suspensions on the polarization of light recorded above the sea surface (see Chami et al.,2005; Chami, 2007). Further studies have also shown the possibility of knowing the composition of the suspension, i.e., the ratio of mineral to organic suspension, using polarization properties of water-leaving radiance (see Gilerson et al., 2006; Chami, 2007; Tonizzo et al., 2011). The intent of the authors was to show the seasonal variability of polarization of the upwelling radiance above the sea surface.

The most challenging part in the analysis of the signal registered by passive radiometric sensors at the top of the atmosphere is to remove the contribution of the reflected photons at the air-sea interface as well as the contribution of the atmosphere. To assess the water-leaving radiance, $L_w(\lambda)$ on the level of accuracy required to derive accurate estimates of the desired water components, other characteristics and methods that support the advanced atmospheric correction and parameter retrieval have been searched, i.e., the black pixel assumption (Siegel et al., 2000), using near infrared and shortwave infrared bands (Wang et al., 2007), using unpolarized ToA reflectance (Frouin et al., 1994), or using polarized water-leaving radiance (Zhai et al. 2017). The latter showed that, in general, the polarized signal at the ToA is 2-3 times higher than its water-leaving part because of the influence of molecular scattering in the atmosphere. Discussions on the use of remote polarization measurements to determine the aerosol properties that can then be used for atmospheric correction are included in Chowdhary et al. (2002), Mishchenko and Travis (1997) as well as Hasekamp and Landgraf (2005). Pust et al. (2011) showed that measurements of the degree of polarization of the sky (made from the ground) can also be helpful in obtaining aerosol parameters. He et al. (2014) proposed to measure the parallel polarization radiance (PPR) instead of radiance intensity at ToA. According to them, such measurements would enhance the OCRS capability. Liu et al. (2017), based on radiative transfer modeling and laboratory measurement, showed that the concentration of particulate matter influences the PPR measured at ToA.

Polarized signal can be measured from the satellite sensors, e.g., POLarization and Directionality of the Earth's Reflectances sensor (POLDER-2), above water using a polarization imaging camera (Freda et al., 2015) or underwater as by Loisel et al. (2008) and Harmel et al. (2008). The measurements are often supported by numerical modeling. Although there are many





ongoing numerical radiative transfer models, very few of them include light polarization (Piskozub and Freda, 2013; Chami et al., 2001). Most current methods of radiative transfer treat light as a scalar to hasten the computations. Polarized radiative transfer was first applied by Chami et al. (2001) to retrieve the polarizing properties of the marine phytoplankton and minerals for different water conditions. Their analysis revealed that the application of the polarization of light in ocean color

algorithms might significantly improve the retrieval of hydrosol properties, especially in coastal waters. Piskozub and Freda (2013) applied their polarized radiative transfer model to the Baltic Sea. They examined how scattering properties of seawater represented by a single scattering albedo affected the polarization of water-leaving radiance. They demonstrated the impact of air bubble layers of various concentrations on the degree of polarization of water-leaving light. They also concluded that polarization remote sensing should be performed on a plane tilted approximately 90° from the solar azimuth

angle to avoid sun glints.

The involvement of polarization in radiative transfer analysis seems to be especially important in coastal areas, knowing that they undergo dynamic changes due to human proximity (Drozdowska et al., 2017), river inflows and the occurrence of pollution, including optically significant oil pollution (Drozdowska et al., 2013). Nevertheless, little attention has been paid to the measurement and modeling of light polarization in coastal areas and closed water basins characterized by optically

complex waters. The Baltic Sea represents a region of a great economic importance, extremely high marine traffic and the impact of inflows from nine different surrounding countries. Inherent optical properties of Baltic seawater and its constituents have been in the spotlight for oceanographers for two decades. In addition to regular measurements of depth profiles of absorption and attenuation coefficients, measurements for different components of seawater have been performed. Colored dissolved organic matter (CDOM) is known to be the primary absorber in the Baltic Sea (Kowalczuk et al., 2010),

and its impact on the total absorption coefficient for blue light can reach up to 80% (Kowalczuk et al., 2005). Kowalczuk (1999) and Kowalczuk and Kaczmarek (1996) found that the high absorption of CDOM in spring and low absorption in winter is due to the biological cycle as well as the seasonal variability of the inlet with river water and its mixing. The search for a better correlation of the spectral remote sensing reflectance $R_{rs}$ ratio with the absorption coefficient of CDOM has been intensively searched within the SatBaltic system (Meler et al., 2016a). Measurements of suspended matter IOPs in the Baltic,

i.e., particle absorption and particle scattering coefficients, have been compared with biogeochemical characteristics of suspended matter such as concentrations of suspended particulate matter, particulate organic matter, particulate organic carbon and chlorophyll a (Woźniak et al., 2011). Meler et al. (2016b) concluded that absorption properties of non-algal particles undergo larger regional than seasonal variability. In addition to the absorption and attenuation coefficients, the volume scattering functions (VSFs) were also measured in the waters of the Southern Baltic (Freda et al., 2007; Freda and

Piskozub, 2007; Freda, 2012). Unique measurements have been performed by the prototype volume scattering meter, characterized by an angular resolution of 0.3º and a range of scattering angles from 0.6º to 177.9º, described by Lee and Lewis (2003). The same instrument has also been used by Chami et al. (2005) in the Black Sea and by Berthon et al. (2007) in the Adriatic Sea. Baltic Sea waters are often affected by small-scale oil pollution (Rudź et al., 2013). The influence of dispersed oil droplets on the absorption coefficient of seawater was researched by Otremba (2007) as well as Haule and





Freda (2016), while their influence on scattering properties has been tested by Freda (2014). The consequences of changes in IOPs for remote detection of dispersed oil pollution have been discussed by Otremba et al. (2013), Otremba (2016), and Haule et al. (2017) based on radiative transfer modeling. Knowledge and datasets collected in the Baltic throughout the past two decades helped us to perform a unique study on polarized radiation above the southern Baltic sea surface. This study

shows the application of a polarized radiative transfer model in three optically different regions, two seasons and two different sea states. The study highlights the possibilities and consequences of including polarization information in bio-optical models of seawater.

## 2 Methods

The difficulty of comparison of upwelling radiance DoP over a wind-roughened southern Baltic surface for various seasons

is caused by the small number of sunny days in winter and too many variable weather factors that would make it difficult to explain the differences. These undesirable weather factors are different aerosol optical depths, sky overcast, and changing speed and direction of wind relative to the position of the sun. For those reasons, we applied a polarized radiative transfer model based on the Monte Carlo code to describe the effect of seasonal changes on the polarization of upwelling radiance. The simulations involved seasonally averaged measurements of inherent optical properties from the Southern Baltic basin

and were run for constant weather conditions. For a detailed description of the inputs and conditions under which the simulation is performed, see the following subsections.

### 2.1 Polarized radiative transfer model - theoretical background

Numerical simulations were carried out using the Monte Carlo algorithm created by Jacek Piskozub and described in Piskozub and Freda (2013). The algorithm solves the vector radiative transfer equation for the atmosphere-ocean system

using the successive orders of scattering method and the Stokes formalism to track the polarization of photons. The algorithm collects information about virtual photons involved statistically in optical processes: reflection at the rough sea surface, refraction at the air-water interface, scattering and absorption within the water body and reaching the ToA. Moreover, the algorithm tracks polarization changes of each photon during these processes. Polarization information is described by four elements of the Stokes vector: $S = [I,Q,U,V]^T$, where I is the total radiance of light, Q describes the

radiance of linearly polarized light (vertical to horizontal), U describes the radiance of linearly polarized light (diagonal right-skewed to left-skewed), V describes the circular polarization (clockwise to counterclockwise) and T denotes the transposition. Three elements (Q, U and V) of the Stokes vector may be both positive or negative. The single quantity that characterizes these properties is the degree of polarization:

$$DoP = \frac{\sqrt{Q^2+U^2+V^2}}{I},$$    (1)

The defined degree of polarization is often replaced by the degree of linear polarization, DoLP.



$$DoLP = \frac{\sqrt{Q^2+U^2}}{I}, \tag{2}$$

The latter hardly differs from DoP because circular polarization represented by the V element of the Stokes vector is negligible in sea water (see off-diagonal Mueller matrix elements in (Voss and Fry, 1984)).

Our polarized radiative transfer model involves a virtual light source to send randomly polarized photons and track their pathways in the means of the probability of occurrence of the processes mentioned above. Reflection and refraction processes are basically described by Fresnel equations, and the slopes of the sea surface are characterized by the wind-dependent distribution of Cox and Munk (1956). The probability of processes within the water body is determined by the corresponding coefficients of absorption and scattering (including multiple scattering). Angular distribution of scattered photons is described by phase functions that, for both atmosphere and sea depth, are characterized separately for molecular scattering and particle scattering. Polarization properties of particle scattering are described by Müeller matrices that for sea water are taken from Voss and Fry (1984) and for atmospheric aerosol particles from Volten et al. (2001). The Müeller matrix and thus the total and polarized phase function of the particles is computed by means of Mie theory. The model outputs the angular distribution of the upwelling radiance and its degree of polarization at any desired level. All results are specified in the principal plane.

## 2.2 Polarized radiative transfer model - input parameters

This section reports the input parameters used in the computations. The data set of the absorption and attenuation coefficients of seawater constituents come from *in situ* measurements in the southern Baltic contained in Sagan (2008). It is the largest dataset of ac-9 measurements in the southern Baltic that was published in a tabular form of average values, extreme values and standard deviations. The instrument was calibrated in ultrapure water and routinely checked for stability with air readings. The standard recommended data processing was performed (Zaneveld et al. 1994). Absolute precision of measurement is 0.005 m$^{-1}$, while relative precision is estimated from 4% in clear waters to 12% in the areas of turbid waters. The data have been recorded during cyclical cruises aboard R/V Oceania in 1999 and 2003 to 2005. Measurements were made in different months of the year. The data set was divided into two seasons, called here 'winter', for the months from November to March and 'summer', the months from April to October. The summer season is characterized by strong phytoplankton growth and the winter season by low biological activity. In addition, Sagan (2008) distinguished three regions: open Baltic, gulfs (Gulf of Gdańsk and Pomeranian Gulf) and coastal waters. The measuring stations, divided into these three types of water, are shown in Figure 1. For defined regions – open Baltic and coastal areas – those two data sets are statistically significantly different for all optical parameters (*t* test for means) at the level p<0.05 by Sagan (2008).

Total absorption coefficient taken to the simulation is a sum $a = (a - a_w) + a_w$, where $(a - a_w)$ is an average absorption coefficient of the N number of measured depth profiles (see Table 1) made with the ac-9 after Sagan (2008), and $a_w$ is the pure water absorption coefficient and comes from Pope and Fry (1997). Similarly, the total attenuation coefficient is defined





as $c = (c - c_w) + a_w + b_w$, where the component $(c - c_w)$ comes directly from Sagan (2008), but to get the total attenuation coefficient, it was enlarged by clean water components of absorption $a_w$ and scattering $b_w$ (Smith and Baker, 1981). According to Sagan (2008), the highest values of IOPs and their highest variability are observed for the water of gulfs and estuaries of rivers that are located nearby. The simulations are carried out at nine wavelengths, namely, 412 nm, 440 nm, 488 nm, 510 nm, 532 nm, 555 nm, 650 nm, 676 nm and 715 nm, which correspond to the ac-9 and are commonly dedicated to ocean color analysis.

Solar zenith angles in the southern Baltic region depend strongly on the season. In months described by Sagan (2008) as the summer season, the highest sun position over the horizon, that means the minimum of SZA during sun culmination, varies between 31º in June (the longest day of the year) and 69º at the end of October. In the modeling, a single value of 45º was chosen as a summer SZA. For months of the winter season, the minimum of SZA varies between 50º in the end of March and 78º in December (the shortest day). Given values are reached at approximately noon and are higher in the rest of the days. That is why SZA of 75º is chosen as a representative for the winter season. Computations are performed for the direction of wind twisted by 45º from the sun reflection plane, chosen arbitrarily. Two wind speeds of 5 m/s and 15 m/s are considered.

Here and in the following figures, the celestial hemisphere and its reflection patterns are represented in a two-dimensional coordinate system. The zenith and the nadir are at the origin and the horizon is represented by the outermost circle. The zenith angle and azimuth angle are measured radially and tangentially, respectively. The solar azimuth angle is always set to 0.

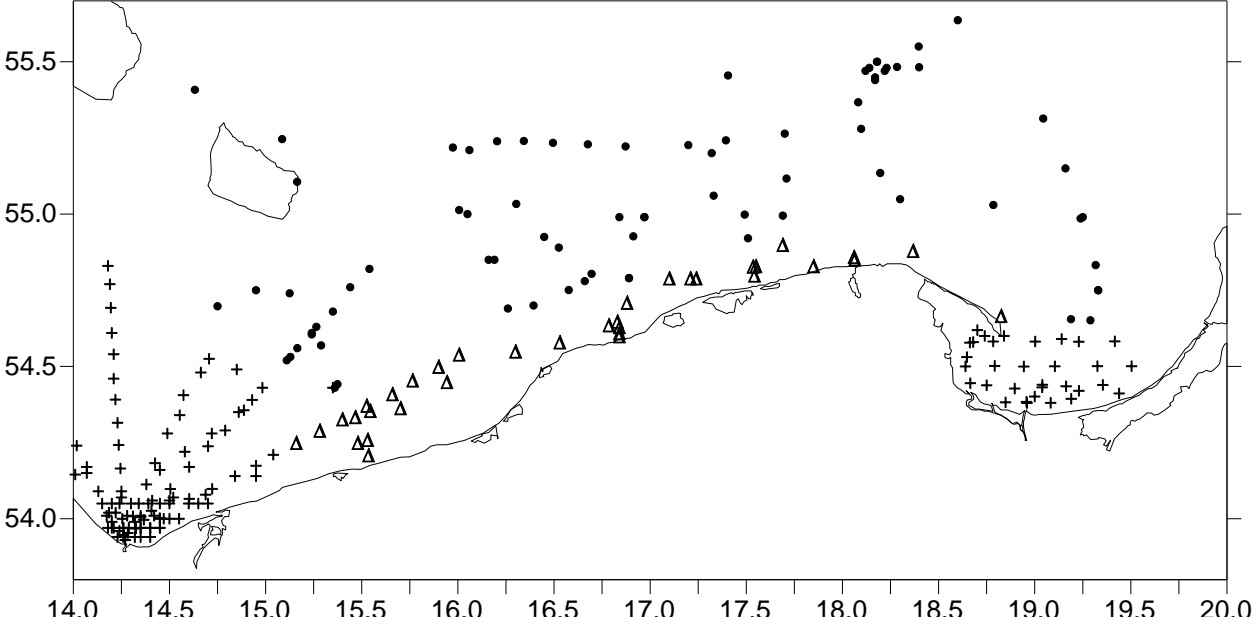

**Figure 1. The area of the southern Baltic Sea, on which the positions of measurement stations are marked, divided into three areas: Pomeranian Gulf and Gulf of Gdańsk (+), coastal water (Δ) and open Baltic water (•).**





**Table 1. Average values of total absorption coefficients, total attenuation coefficients, their standard deviations and their ratios. All values measured by Sagan (2008) in southern Baltic in 1999 and 2003 to 2005 are averaged for depths 0 to 5 meters and then averaged for N measuring stations**

| Summer Season | | | | | | | | | | | | | | | |
|---|---|---|---|---|---|---|---|---|---|---|---|---|---|---|---|
| $\lambda$ | Baltic, N=930 | | | | | Gulfs, N = 1428 | | | | | Coastal Waters, N = 132 | | | | |
| [nm] | $a(\lambda)$ | $\sigma_a$ | $c(\lambda)$ | $\sigma_c$ | $\frac{a(\lambda)}{c(\lambda)}$ | $a(\lambda)$ | $\sigma_a$ | $c(\lambda)$ | $\sigma_c$ | $\frac{a(\lambda)}{c(\lambda)}$ | $a(\lambda)$ | $\sigma_a$ | $c(\lambda)$ | $\sigma_c$ | $\frac{a(\lambda)}{c(\lambda)}$ |
| 412 | 0.595 | 0.12 | 1.230 | 0.40 | 0.483 | 1.095 | 0.65 | 2.560 | 1.64 | 0.428 | 0.615 | 0.12 | 1.470 | 0.52 | 0.418 |
| 440 | 0.396 | 0.10 | 0.999 | 0.37 | 0.396 | 0.786 | 0.52 | 2.209 | 1.48 | 0.356 | 0.416 | 0.10 | 1.249 | 0.52 | 0.333 |
| 488 | 0.214 | 0.07 | 0.817 | 0.34 | 0.262 | 0.444 | 0.32 | 1.887 | 1.35 | 0.236 | 0.214 | 0.06 | 1.027 | 0.46 | 0.209 |
| 510 | 0.183 | 0.05 | 0.785 | 0.33 | 0.233 | 0.353 | 0.23 | 1.825 | 1.31 | 0.193 | 0.173 | 0.04 | 0.995 | 0.46 | 0.173 |
| 532 | 0.155 | 0.04 | 0.756 | 0.32 | 0.205 | 0.285 | 0.18 | 1.756 | 1.28 | 0.162 | 0.155 | 0.03 | 0.946 | 0.44 | 0.163 |
| 555 | 0.140 | 0.03 | 0.731 | 0.31 | 0.192 | 0.240 | 0.13 | 1.711 | 1.26 | 0.140 | 0.140 | 0.03 | 0.911 | 0.42 | 0.153 |
| 650 | 0.380 | 0.02 | 0.921 | 0.29 | 0.413 | 0.430 | 0.08 | 1.811 | 1.20 | 0.237 | 0.380 | 0.02 | 1.071 | 0.38 | 0.355 |
| 676 | 0.518 | 0.05 | 1.029 | 0.28 | 0.503 | 0.638 | 0.19 | 1.909 | 1.18 | 0.334 | 0.508 | 0.03 | 1.169 | 0.38 | 0.435 |
| Winter Season | | | | | | | | | | | | | | | |
| $\lambda$ | Baltic, N = 234 | | | | | Gulfs, N = 540 | | | | | Coastal Waters, N = 36 | | | | |
| [nm] | $a(\lambda)$ | $\sigma_a$ | $c(\lambda)$ | $\sigma_c$ | $\frac{a(\lambda)}{c(\lambda)}$ | $a(\lambda)$ | $\sigma_a$ | $c(\lambda)$ | $\sigma_c$ | $\frac{a(\lambda)}{c(\lambda)}$ | $a(\lambda)$ | $\sigma_a$ | $c(\lambda)$ | $\sigma_c$ | $\frac{a(\lambda)}{c(\lambda)}$ |
| 412 | 0.485 | 0.03 | 0.680 | 0.015 | 0.713 | 0.755 | 0.31 | 1.760 | 1.04 | 0.429 | 0.585 | 0.14 | 1.250 | 0.68 | 0.468 |
| 440 | 0.296 | 0.02 | 0.479 | 0.015 | 0.618 | 0.496 | 0.22 | 1.469 | 0.95 | 0.338 | 0.376 | 0.10 | 1.009 | 0.63 | 0.373 |
| 488 | 0.154 | 0.02 | 0.337 | 0.015 | 0.459 | 0.264 | 0.13 | 1.217 | 0.86 | 0.217 | 0.194 | 0.06 | 0.817 | 0.57 | 0.238 |
| 510 | 0.133 | 0.02 | 0.315 | 0.014 | 0.421 | 0.213 | 0.10 | 1.165 | 0.83 | 0.182 | 0.163 | 0.05 | 0.785 | 0.55 | 0.207 |
| 532 | 0.125 | 0.02 | 0.296 | 0.013 | 0.421 | 0.185 | 0.08 | 1.106 | 0.81 | 0.167 | 0.145 | 0.04 | 0.756 | 0.54 | 0.191 |
| 555 | 0.120 | 0.01 | 0.291 | 0.013 | 0.411 | 0.170 | 0.06 | 1.071 | 0.79 | 0.158 | 0.140 | 0.03 | 0.731 | 0.52 | 0.191 |
| 650 | 0.370 | 0.01 | 0.521 | 0.012 | 0.711 | 0.400 | 0.04 | 1.211 | 0.73 | 0.330 | 0.370 | 0.01 | 0.931 | 0.46 | 0.398 |
| 676 | 0.478 | 0.02 | 0.629 | 0.012 | 0.760 | 0.528 | 0.08 | 1.299 | 0.72 | 0.407 | 0.488 | 0.02 | 1.039 | 0.45 | 0.470 |

5 ## 3 Results and discussion

### 3.1 Extreme values of the degree of polarization

Examples of simulation results are presented in Figures 2a and 2b in the form of polar plots of the degree of polarization of upwelling radiance just above the sea surface. Figure 2a shows the DoP for the average IOPs measured in the open waters of the Baltic Sea for a wavelength of 412 nm in the summer season, while Figure 2b depicts an analogous case for the winter

10 season. These two plots are characterized by one of the highest values of the peak of DoP of 0.88 for summer and 0.84 for



winter. The azimuth position of the sun is 0° in all cases. Corresponding values of computed upwelling irradiance I (in units of $Wm^{-2}sr^{-1}nm^{-1}$) are shown on the plots of Figures 2c and 2d on the logarithmic scale (due to their high angular variability).

**Figure 2.** Simulation results of above-water upwelling radiance for average IOPs of open waters of the southern Baltic, wavelength 412 nm: a) DoP in the summer season, SZA 45°, b) DoP in the winter season, SZA 75°, c) decimal logarithm of upwelling radiance in the summer season, SZA 45°, d) decimal logarithm of upwelling radiance in the winter season, SZA 75°.

The small SZA of the summer season (45°) resulted in low values of upwelling radiance that are stretched from the direct reflection point to the horizon, where it is extended both left and right from the azimuth of 180°. The high SZA of the winter season (75°) resulted in much higher values of reflected light, which are also stretched from reflection point to the horizon.



Examples of the lowest values of the maximum DoP, referred to as max(DoP), are shown in Figures 3a and 3b. They were obtained for the regions of Gulf of Gdańsk and Pomeranian Gulf, simulated for the spectral band of 555 nm, for the summer season (Figure 3a) and for the winter season (Figure 3b).

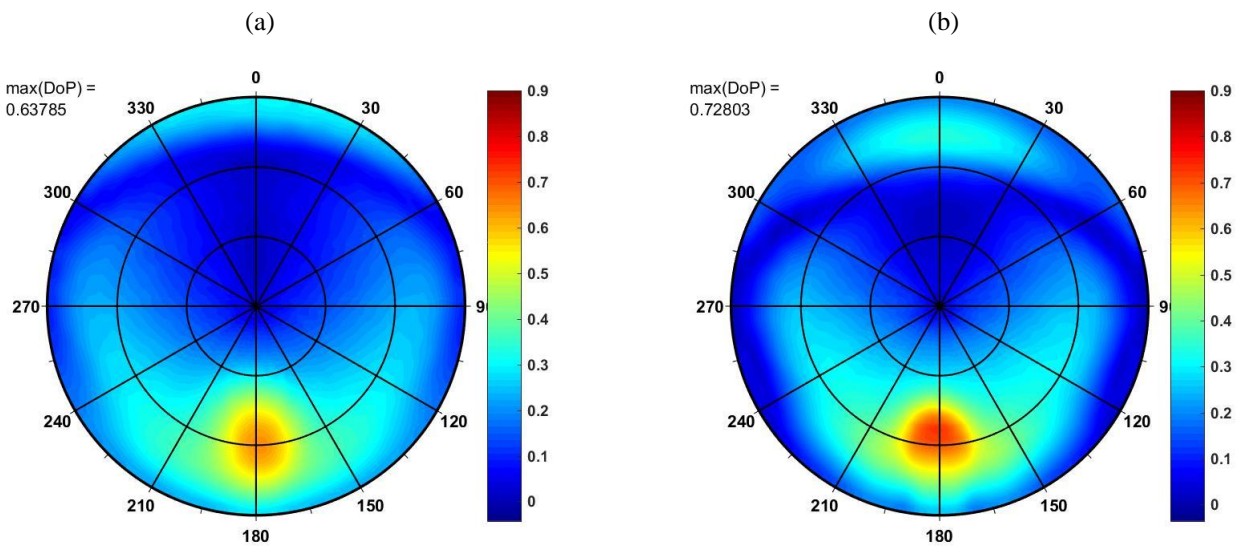

**Figure 3. Simulation results of above-water upwelling radiance for average IOPs of gulf waters of the southern Baltic, wavelength 555 nm: a) DoP in the summer season, SZA 45°, b) DoP in the winter season, SZA 75°.**

The maximum values of DoP presented in Figure 2a (summer season, open Baltic waters) are visible for azimuth angles close to 180º (direction of reflected sun) and a zenith angle of approximately 55º although the solar zenith angle is 45º, while the max(DoP) in Figure 3a (summer season, waters of gulfs) is visible for zenith angle of approximately 60º. In contrast to the summer season case, the maximum DoPs in the winter season are close to the zenith angle of 48º (Figure 2b) and 54º (Figure 3b), while SZA is 75º. The lower position of the sun results in a higher position of the maximum DoP of upwelling radiation than its reflection angle, and a higher position of the sun results in a lower position of the maximum DoP. Another interesting effect is the higher DoP observed for directions close to the incident rays of the sun (azimuth 0º). In general, in the winter season, the values of DoP are higher than in summer, and zenith angles of this effect are lower in the winter than in the summer season.

### 3.2 Spectral variability of the degree of polarization

The results of Monte Carlo simulations of angular characteristics of DoP of upwelling radiance are presented in Figure 4. These results are obtained for average IOPs of open Baltic waters for wind speed of 5 m/s for three wavelengths (440 nm, 555 nm, 650 nm) and for both seasons. Vertical cross-sections of such polar plots for the same type of water (open Baltic

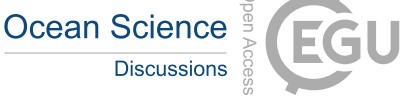



Sea) but additionally for all examined wavelengths and for two speeds of wind, 5 m/s and 15 m/s, are presented in Figure 5. Such cross-sections show the DoP in the principal plane, including the direction of incident sun beam (on the left side of the plot), zenith and direction of sun reflection beam for calm sea surface (on the right side). The azimuth direction of the sun position, described as 0° in the polar plots, is marked by negative zenith angles in Figure 5, while azimuth directions of 180° that include the sun reflection beam are marked by positive zenith angles.

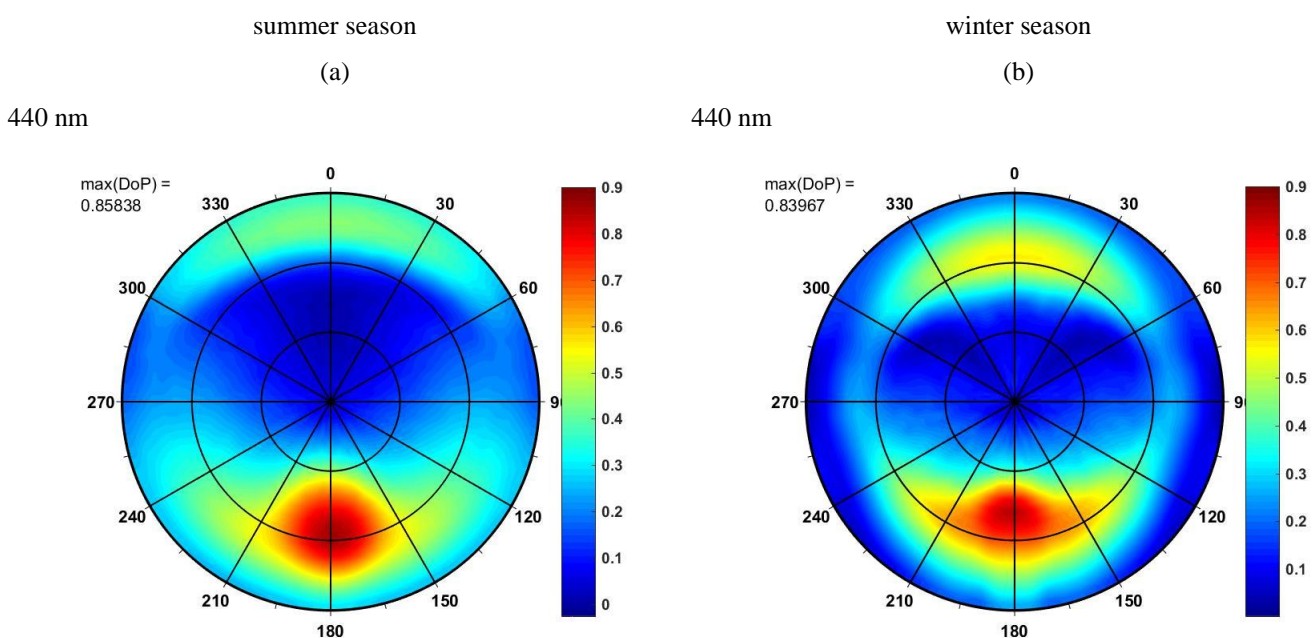





**Figure 4. Simulation results of above-water upwelling radiance for average IOPs of open Baltic Sea water, for speed of wind of 5 m/s: a) DoP in the summer season, λ=440 nm, SZA 45°, b) DoP in the winter season, λ=440 nm, SZA 75°, c) DoP in the summer season, λ=555 nm, SZA 45°, d) DoP in the winter season, λ=555 nm, SZA 75°, e) DoP in the summer season, λ=650 nm, SZA 45°, f) DoP in the winter season, λ=650 nm, SZA 75°.**





summer season

winter season

(a)

(b)

(c)

(d)

**Figure 5. Degree of polarization plotted for the principal plane, e.g., plane containing both incident ray of the sun and zenith direction (cross section through polar plots for azimuths 0° and 180°). Azimuth 0° - azimuth of sun position is marked by negative zenith angles, while azimuth 180° - that contains sun reflection is marked by positive zenith angles. Simulation results for open Baltic Sea water: a) summer season, SZA 45°, wind speed 5 m/s, b) winter season, SZA 75°, wind speed 5 m/s, c) summer season, SZA 45°, wind speed 15 m/s, d) winter season, SZA 75°, wind speed 15 m/s.**

The analysis of individual spectral bands shows that high values of DoP correspond to the high absorption-to-attenuation ratio for the total of visible light domain (see Table 1). High values of absorption coefficient for 650-676 nm wavelengths (in the red spectral region) are caused by pure water (see Pope and Fry, 1997) while high absorption coefficients for





wavelengths of the blue-green range are caused mainly by CDOM (Kowalczuk et al., 2005). The lowest values of max(DoP) for each type of water and for each season are observed for the 555 nm spectral band. The lowest values of absorption and weak spectral variability of the scattering coefficient implies that the wavelength of 555 nm is characterized by the lowest absorption-to-attenuation ratios due to the existence of a minimum of absorption for seawater containing phytoplankton.

Algae cells, depending on the composition of their pigments, may have a minimum of absorption in a wide range of spectral bands from 550 nm to 660 nm (Bricaud et al., 2004). Considering the absorption of pure water that is increasing with wavelength (Pope and Fry, 1997), the minimum of the absorption in Baltic waters for the spectral band of 555 nm results.

The spectral shape of the DoP cross-sections contains two maxima, and their angular positions depend on the absorption-to-attenuation ratio, the season and the wind speed. The angular position of the higher maximum depends mostly on the season,

varying from approximately 60º in the summer to 35º-50º in the winter (see Figure 5). The lower maximum we observe at the zenith angles between -70º and -90º in the summer as well as between -55º and -70º in the winter. Higher wind speed of 15 m/s, in comparison to 5 m/s, causes the irregular shape of peaks. Moreover, the higher wind speed causes an increase of the DoP value for a lower maximum in the 650 nm and 676 nm spectral bands and its shift to a higher position (toward the zenith). At the same time, the DoP values for shorter wavelength bands are decreased and shifted to a lower position (toward

horizon).

### 3.3 Regional variability of the degree of polarization

Computations of DoP were carried out in three optically different regions of the southern Baltic. Such a division is justified in previous studies of optical and hydrological properties of the south Baltic waters (Olszewski et al., 1992). They showed a relationship between the measured values of IOPs and their location in relation to river estuaries, distance from the shore or

bathymetry of the bottom.

Comparison of water type influence on the DoP is shown in Figure 6 for two wavelengths, 440 nm (Figure 6a) and 555 nm (Figure 6b). The type of water has less influence on the DoP than the season and its representative SZA.



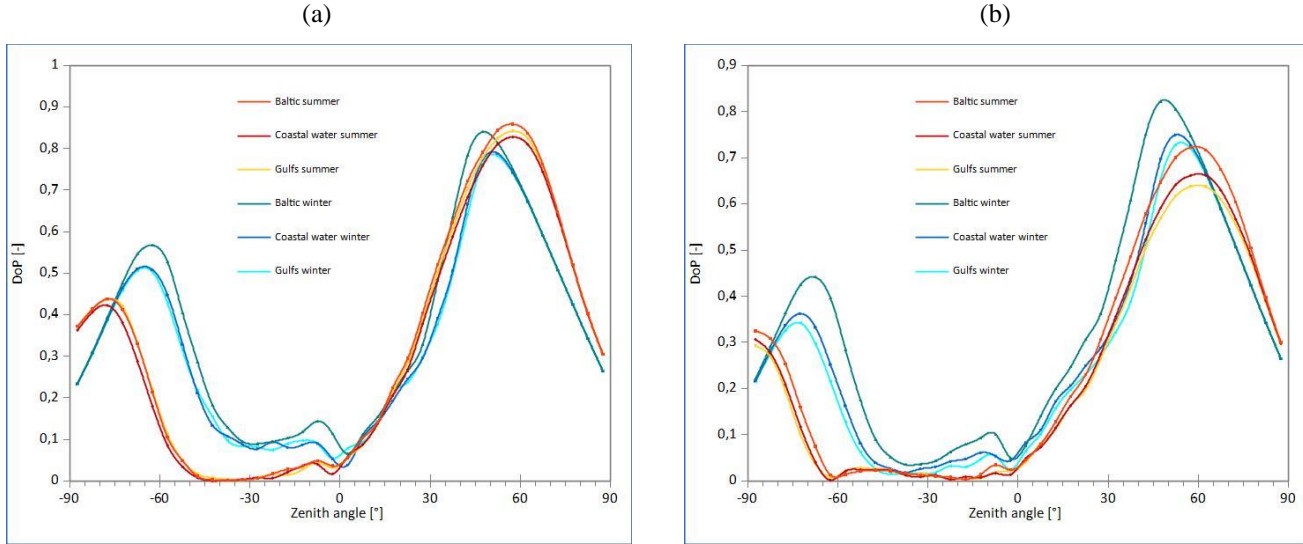

**Figure 6. Degree of polarization plotted for the principal plane, i.e., plane containing both incident ray of the sun and zenith direction (cross-section through polar plots for azimuths 0° and 180°). Azimuth 0° is marked by negative zenith angles. Simulation results for wind speed of 5 m/s for all types of water and two seasons: a) 440 nm, and b) 555 nm.**

However, the highest values of DoP for most zenith angles and the highest values of its peak max(DoP) for each season are observed for open Baltic Sea water. Coastal waters and gulfs are characterized by similar values of DoP in each season. For the wavelength of 440 nm, in the summer, the differences of max(DoP) between the open Baltic and other regions reach 0.02-0.04, while in the winter those differences exceed 0.05. For the 555 nm band, in the summer the differences of max(DoP) between the open Baltic and other regions reach 0.06-0.09 and in the winter these differences reach 0.07-0.09, respectively. We also observed another regional difference in the angular position of the maxima of DoP that is noticeable in the winter season only. Two maxima of the DoP cross-sections are closer for open Baltic waters than for gulfs and coastal waters.

In the following section, we explain that the degree of polarization depends on the absorption-to-attenuation ratio, and all its regional changes are the result of the absorption-to-attenuation ratio variability.

### 3.4 DoP dependence on the absorption-to-attenuation ratio

This section contains the comparison of the degree of polarization for summer and winter seasons as a function of the absorption-to-attenuation ratio.

The total $a(\lambda)/c(\lambda)$ ratio (see Table 1) is higher in the open Baltic water than in other regions because of the low scattering coefficients (Sagan, 2008). The value of the latter is determined mainly by the concentration of suspended matter, which in open waters is significantly lower than in gulfs or coastal/near-shore waters. According to Sagan (2008), the average particle scattering coefficient does not depend strongly on wavelengths, and in open Baltic waters in the winter season it varies between 0.15 (for 676 nm) and 0.19 (for 412 nm). In the same season, but in the waters of gulfs, the average particle





scattering coefficient varies between 0.77 (for 676 nm) and 1.00 (for 412 nm). For the influence of water type on DoP, regardless of the wavelength and wind speed, the lowest max(DoP) values in winter are observed in the waters of gulfs. These waters are characterized by the highest scattering coefficients because of the high inflow of particulate matter with river waters. However, in the summer season, the lowest peak of DoP is observed for spectral bands from 412 nm to 532 nm

in coastal waters, and in wavelengths from 555 nm to 676 nm in gulfs. The values of $a(\lambda)/c(\lambda)$ for these types of water differ by less than 5% except for the 650 nm and 676 nm spectral bands. According to the Fresnel equations, the reflections of two polarized components (parallel and perpendicular to the transmission/reflection plane) from the sea surface depend on two parameters only. These parameters are the relative refractive index of the medium and the angle of incidence of the light beam. Thus, polarization of the reflected light does not depend on the IOPs of seawater. The observed differences in

polarization of upwelling radiance above the sea surface for various absorption and attenuation coefficients of seawater come from the water-leaving component of that radiance but not from the reflected part.

All the values of maximum DoP of above-water upwelling radiance obtained for each absorption-to-attenuation ratio are collected in Figure 7. The summer season case is depicted in Figure 7a while the winter case is depicted in Figure 7b. The water types are marked with different symbols, and two wind speeds are marked with different colors. The correlations of

the max(DoP) to the ratio of $a(\lambda)/c(\lambda)$ are approximately linear. However, correlation coefficient analysis has shown that the power functions are better matched. The reason for the nonlinearity of this correlation may be related to the potential obtaining or even exceeding one by the value of DoP for certain combinations of absorption and attenuation coefficients. The trend line for the plot depicted in Figure 7a shows the relationship of the maximum of DoP to the ratio of $a(\lambda)/c(\lambda)$ for the summer season and in Figure 7b for the winter season, respectively. The trend lines presented may be described by

power functions:

$$max(DoP) = A \left( \frac{a(\lambda)}{c(\lambda)} \right)^{B}, \tag{3}$$

whose parameters are collected in Table 2. These correlations are obtained for various spectral channels. Hence, they are wavelength-independent for the examined visible spectral range.

**Table 2. Parameters of equation (3), which describes the power trend lines in Figures 4 (a) and 4 (b)**

| Simulation Conditions | A | B | $R^2$ |
|---|---|---|---|
| SZA 45°, wind speed 5 m/s | 1.102 | 0.262 | 0.973 |
| SZA 45°, wind speed 15 m/s | 0.997 | 0.250 | 0.996 |
| SZA 75°, wind speed 5 m/s | 0.903 | 0.117 | 0.906 |
| SZA 75°, wind speed 15 m/s | 0.914 | 0.173 | 0.990 |





(a)                                                           (b)

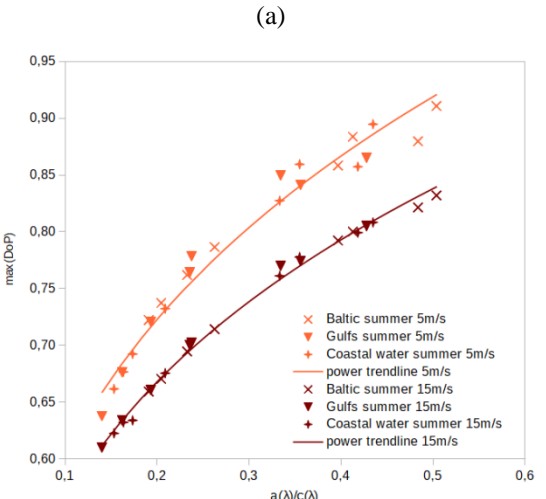     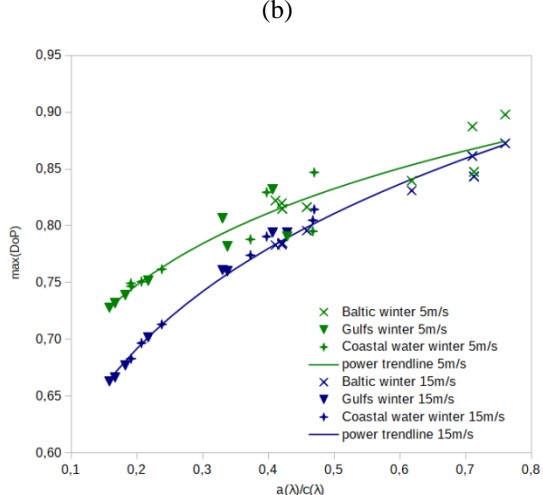

**Figure 7. Values of maximum of degree of polarization against absorption to attenuation ratios a(λ)/c(λ) for average values of IOPs presented in Table 1, plotted for (a) the summer season and (b) the winter season.**

An analysis of all collected data shows that higher values of maximum DoP are observed for lower wind speed. Moreover, max(DoP) has a higher range of variability in the summer season than in the winter. Figure 7a (summer season) shows that

values of max(DoP) are between 0.64 and 0.91 for the wind speed of 5 m/s, and between 0.61 and 0.83 for the wind speed of 15 m/s. However, in Figure 7b (winter season), the values are between 0.73 and 0.9 for the wind speed of 5 m/s, and between 0.66 and 0.87 for the wind speed of 15 m/s. Power trend lines for the same wind speed for summer and winter seasons (in Figures 7a and 7b) intersect. For a wind speed of 5 m/s, by a(λ)/c(λ) equal to 0.26, both power functions reach the same max(DoP) of 0.77. For a wind speed of 15 m/s, by a(λ)/c(λ) equal to 0.32, both power functions reach the same value of

0.75. For lower absorption-to-attenuation ratios, winter DoPs have higher values than summer and for higher a(λ)/c(λ) values, the summer DoPs are higher.

The reason for the correlation of the maximum DoP with the absorption-to-attenuation ratio is the occurrence of multiple scattering in water depth. The degree of polarization decreases after each act of scattering on particle matter. A high absorption-to-attenuation ratio means simply low scattering-to-attenuation impact and, hence, shallow penetration of light in

the water column and low participation in multiple scattering that decreases the DoP. Such conclusion is in accordance with Piskozub and Freda (2013), who examined the influence of single scattering albedo on the polarization of water-leaving radiance. Their results show that in the sun reflection plane, the highest value of DoP is observed when the total scattering coefficient is the lowest (see Figure 3 in Piskozub and Freda (2013)).

The influence of wind speed on the DoP values shown in Figures 7a and 7b is very clear: sea surface roughness depolarizes

the reflected light. Zhou et al. (2013) demonstrated that wind speed and wind direction can change the polarization patterns of reflected skylight from a rough sea surface to a certain extent. Our study shows, in particular, that high wind speed results in lower values of max(DoP) of the total upwelling radiance. Such regularity is filled for all types of water and all spectral bands.




## 3.5 Direction of the highest polarization peak

Moreover, as an attempt to explain the differences between seasons, the DoPs of underwater upwelling radiances were simulated. The results of the computations are shown in Figures 8a and 8b, and the corresponding values of the radiance field are presented in Figures 8c and 8d. These plots indicate that directions characterized by high values of DoP create a dispersed ring, which is tilted from the horizontal direction. The angular tilt of the ring plane is approximately 60º for the summer season and 45º for the winter. Moreover, both the underwater upwelling spectral radiance and its DoP are higher in the summer season than in the winter season.

a)

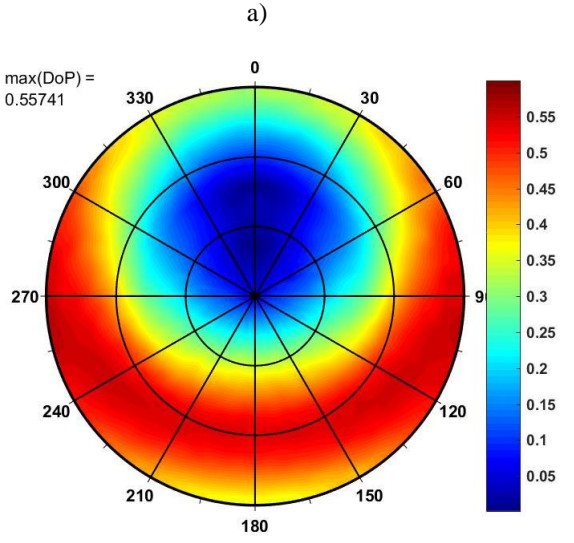

(b)

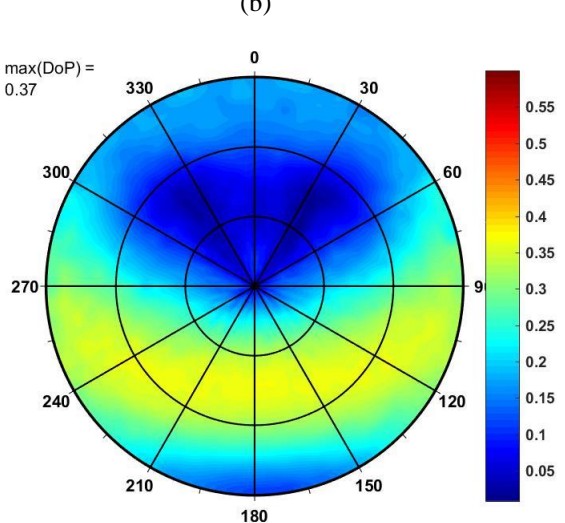



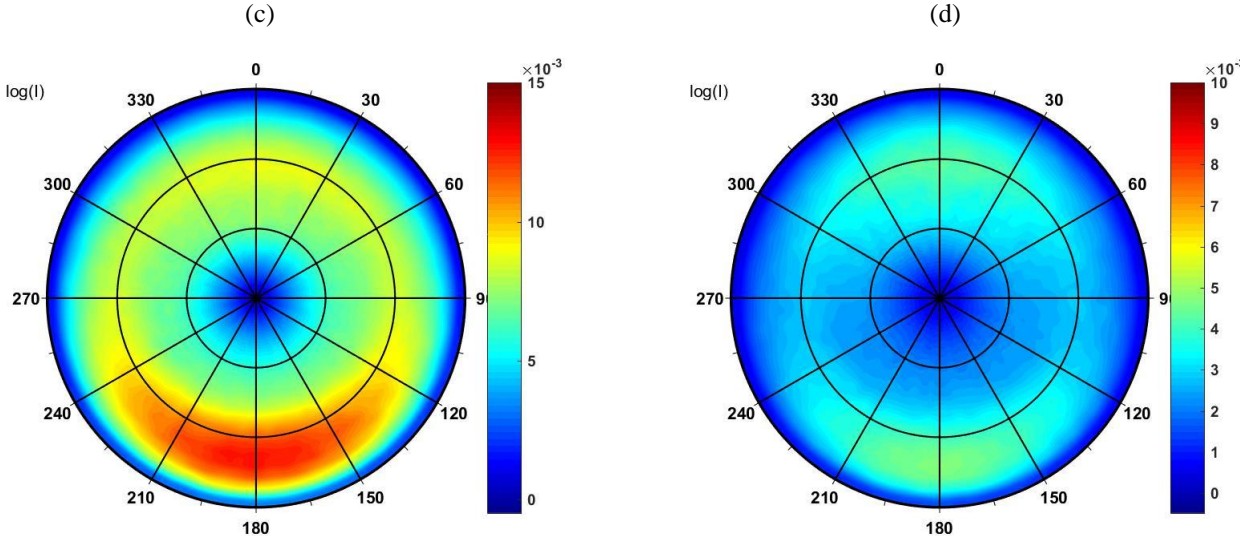

**Figure 8. Simulation results of underwater upwelling radiance for average IOPs of open waters of the southern Baltic, wavelength 412 nm: a) DoP in the summer season, SZA 45°, b) DoP in the winter season, SZA 75°, c) decimal logarithm of upwelling radiance in the summer season, SZA 45°, d) decimal logarithm of upwelling radiance in the winter season, SZA 75°.**

The angular position of highly polarized radiance for both below and above the sea surface is schematically shown in Figure 9. For a plane including sun reflection and refraction (azimuth 0º and 180º) if the surface is calm, then for the SZA of 45°, the transmission angle TA, according to Snell's law, is 32°. Underwater polarization, caused by molecular scattering, is the highest in the plane perpendicular to the direction of propagation. In the vertical plane including transmitted rays, the highest

10 polarization of molecular scattering (HPM) is 58º. For SZA of 75°, the transmission angle is 46º and direction of HPM is 44º. Directions of HPM predicted for a calm sea surface proved to be the same for a wind-roughened sea surface (see the polarization patterns in Figures 2a and 2b). The angular position of HPM, which is perpendicular to TA in the reflection-refraction plane, is both predicted by theory (molecular scattering) and confirmed by the underwater measurements of Tonizzo et al. (2009). Their underwater DoPs, measured in waters characterized by various optical properties and a depth of

15 one meter, slightly exceed the value of 0.4 by the detection angle close to 100º (see their Figure 9), while our simulated underwater values of DoP reach even 0.55 (Figure 8a). The difference may be because the measurements of Tonizzo et al. were performed at a depth of 1 m, and our simulation is obtained for depths just below 0 m.





**Figure 9. Diagram of directions of High Polarization radiance for surface Reflections (HPR) and High Polarization for Molecular scattering (HPM)**

The highest polarization of reflected radiance (HPR), according to Fresnel equations for a flat reflective surface, should appear when the reflection angle is perpendicular to TA. Figures 2a and 2b show that for a rough sea surface, the HPR direction is not equal to the angle of incidence (SZA). For SZA of 45º (summer season case), the HPR is dispersed approximately 56º (see Figure 2a) while for a calm surface it would be exactly 58º. Moreover, the sum of SZA and zenith angle of HPR is 101º, which is very close to double the Brewster angle, that for water is approximately 53º. That could

explain the angular position of HPR. However, for SZA of 75º (representing the winter season in the southern Baltic), the HPR is dispersed around the zenith angle of 48º (see Figure 2b). Here, the direction perpendicular to TA is 44º (see Figure 8b). The sum of SZA and HPR is 123º, which is much higher than double the Brewster angle. For this reason, the position of the HPR for rough sea surface should be interpreted rather as the direction perpendicular to the average TA, than the reflection from these surface slopes for which the angles of incidence and reflection are both equal to the Brewster angle.

The unpolarized sunlight after reflection or refraction processes at the sea surface becomes partly polarized. As an attempt to explain high differences between values of upwelling radiance over sea surface shown in the Figures 2c and 2d, we applied Fresnel equations to calculate both the transmission and reflection fractions of two polarization components of that light, namely, the parallel and the perpendicular. These fractions depend strongly on the solar zenith angle of incident light. For a

calm surface, SZA of 45º (corresponding to the summer season) and TA of 32º, only 0.3% of the parallel polarized light component is reflected by the surface, and 99.7% is transmitted to the water depth, while for the perpendicular polarized part much more (5.5%) is reflected and 94.5% is transmitted through the surface. For an SZA of 75º (winter season) and a TA of 46°, the reflected fraction would be 11% for the parallel polarization component and as much as 32% for the perpendicular component, while the transmission would be equal to 89% and 68%, respectively (see Table 3). These calculations show that

for SZA of 45º the reflected part of the upwelling radiance is much lower than for an SZA of 75º, and the contribution of the perpendicular component of polarization is much higher than the parallel, resulting in a higher degree of polarization of the



reflected part of the total upwelling radiance in the summer season because of higher disproportion of one polarization component.

**Table 3. The contribution to the processes of reflection and transmission through flat sea surface, calculated according to Fresnel equations for two polarization components of unpolarized light coming from SZA. These polarization components are perpendicular and parallel to the plane of reflection-refraction.**

| Summer Season, SZA = 45° | | | |
|---|---|---|---|
| Parallel | | Perpendicular | |
| Reflection | Transmission | Reflection | Transmission |
| 0.3% | 99.7% | 5.5% | 94.5% |
| Winter Season, SZA = 75° | | | |
| Parallel | | Perpendicular | |
| Reflection | Transmission | Reflection | Transmission |
| 11% | 89% | 32% | 68% |

Polarization patterns of total upwelling radiance, combined with radiance reflected off the sea surface and water-leaving radiance, differ from those describing only the reflected part obtained by Zhou et al. (2013). As the reflected part of the upwelling radiance forms 90% and more of the total upwelling radiance during sunny days, the contribution of water-leaving radiance to the polarization changes is detectable. Zhou et al. (2013) show that the DoP of the reflected radiance component creates a ring of high values at the polar plot called the Brewster zone. The angular shape of this zone depends strongly on the SZA (see Figure 10 in Zhou et al. (2013)) and the polarization direction in this area is almost perpendicular to the local meridian. In our study, low transmission of light into the water depth in the winter case (SZA of 75°) in comparison to the summer case (SZA of 45°) also explains the differences in underwater upwelling radiances (see Figures 8c and 8d). Underwater radiance and its polarization have a higher influence on polarization of upwelling radiance over the sea surface in the summer than in the winter. Lower SZA causes higher transmission into the water depths and higher scattering toward the sea surface, so DoP distribution of upwelling radiance in winter looks more similar to a polarization pattern of the reflected part presented by Zhou et al. (2013) with their ring-shaped Brewster zone.

The results of our simulations are in qualitative agreement with the measurements of above water DoLP of the total upwelling radiance presented by Freda et al. (2015). This agreement is the similarity of the peak of degree of polarization on the polar plots, which are stretched along the azimuth angles. Freda et al. (2015) obtained lower values of measured DoLP with a maximum of 30-40% (see Figures 1 and 2 in (Freda et al., 2015)), which is presumably caused by different weather conditions and unknown environmental parameters during measurements, such as a high absorption coefficient in the waters of the river mouth, different aerosol optical depth, or other parameters. However, despite the differences in the maximum



degree of polarization, the angular distribution patterns are similar, with the peak in the vicinity of the sun reflection azimuth angle.

The results of the correlation of the maximum DoP with the absorption-to-attenuation ratio seem to be coincident with the results of Ibrahim et al., (2012) who studied the degree of linear polarization just below the air-water interface. Their

correlation of attenuation-to-absorption ratio with DoLP displays a hyperbolic-like shape (see Figures 5 to 8 in Ibrahim et al. 2012). Therefore, for an inverted ratio of absorption-to-attenuation, it would be near-linear. The modeling results of Ibrahim et al. cannot be compared directly to the results presented in this paper because they received DoLP just below the sea surface, and we focused on DoP just above the surface. However, our choice of seawater absorption-to-attenuation ratio, which can be called the relative absorption value (to total attenuation), as a parameter correlated to degree of polarization

seems to be more suitable.

## 5 Conclusions

In this paper, we have investigated the relationship between the seawater absorption-to-attenuation ratio and the degree of radiance polarization above the rough sea surface. Using a Monte Carlo polarized radiative transfer model, we compared/analyzed simulated polarization patterns in three optically different regions in the southern Baltic (i.e., open

Baltic, gulfs, coastal waters), two seasons (defined by their typical solar zenith angles: 45º for summer and 75º for winter), and two wind speeds of 5 m/s and 15 m/s, each for nine visible spectral bands (412 nm, 440 nm, 488 nm, 510 nm, 532 nm, 555 nm, 650 nm, 676 nm, 715 nm). The use of the modeling tool allowed us to exclude unwanted and unpredictable variables (such as weather conditions and aerosol optical thickness) and to conduct undisturbed comparison of the DoPs of combined water-leaving and reflected components of upwelling radiance.

We found that the variability of the maximum of DoP depends more on seasonal than regional changes and can be explained to a large degree by the absorption-to-attenuation ratio. A thorough analysis has shown that there is a strong correlation between max(DoP) and the ratio mentioned previously. The correlation is well described ($R^2$>0.90) by a power function with factor A close to one and factor B depending more on SZA than on the wind speed. In our study, seasonal variability of the degree of polarization is higher/more significant than regional variability. However, this may be true only in the southern

Baltic region due to the characteristically different SZA ranges in the winter and summer seasons.

Further supplementary simulations of underwater polarization distribution have shown that the directions of max(DoP) of combined water-leaving and reflected components of upwelling radiance are dispersed in directions perpendicular to the undersurface transmission angle.

For the ocean color remote sensing application, only the water-leaving part of the upwelling radiance carries useful

information about bio-optical parameters of seawater, although it is a small fraction of the total upwelling radiance. Polarized radiative transfer modeling makes it possible to separate the water-leaving part and, in this case, noise-inducing reflected part and therefore to enhance the quality of information on the seawater optically active components retrieved by



above-water sensors – airborne or satellites. Our study is a step toward inclusion of polarization properties in the bio-optical models in the Baltic Sea. However, the conclusions from the research, in our opinion, should be universal and apply also to other water bodies.

**Acknowledgments.** The authors are grateful to Prof. Jacek Piskozub for his valuable comments and suggestions. The research presented in this paper was supported by grant No. UMO-2012/07/D/ST10/02865, funded by the National Science Centre (NCN) of Poland and by Gdynia Maritime University statutory research No. DS/427/2018.

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
