# Peer review of "On the role of the seawater absorption to attenuation ratio in the radiance polarization above the Southern Baltic surface"

_Ocean Science, 2018_

## Referee Comment (RC1) · Anonymous Referee #1 · 24 Jan 2019

General Comments

The authors of the manuscript have conducted large number of polarimetric simulations of water conditions in the Baltic sea, over two seasons, and three different water types. An analysis is conducted which looks at the relationship between the absorption to attenuation of the water, and the upwelling polarization signal. It is determined that a correlation exists, and that it also depends on the Sun angle and the wind speed. A discussion about the direction of maximum DoP is also given.

The authors should be commended for the large amount of work that obviously went into this analysis, and especially for the inclusion of light polarization, which many

scientists avoid.

That being said, however, I believe there to be several fundamental scientific problems with this manuscript that must be addressed before it is suitable for publication. I wish to convey to the authors that, although I have written many comments, it is because I am both familiar and passionate about the subject, and wish to see it properly conducted and thereby make an impactful contribution to the field.

My critique may be distilled down to a few main issues:

1. I found the literature cited by the manuscript very lacking. Almost all citations are prior to ~2012, and there has been many advances in the field in recent years.

2. The analysis focuses on "max(DoP)". The maximum DoP is almost always either in, or near, the specular reflection point for above water simulations due to the inherent Fresnel reflectivity of the mean sea surface. (Figs 2,3,4,5,6 of this manuscript) This is an infeasible place to be measuring polarization for ocean color, since any measured signal from the ocean will be overwhelmed by the reflectance of the Sun. Ocean color satellites will not measure at this geometry. The paper would be much more applicable if the max(DoLP) were limited to feasibly measurable angles.

3. I believe that the contribution made by section 3.5 of the manuscript is marginal at best. Most of the conclusions about the direction of max(DoP) are 'known', or can be determined easily from Snell's law and the knowledge that the maximum DoP occurs at scattering angles near 90 degrees. The underwater simulations are illustrative, but the explanations given for the direction of the DoP are inaccurate. See the specific comments below for further details.

Specific Comments

1. Pg 2 line 1-2: Garaba and Zielinkski, 2013 have very little to say about the polarization of above surface light. Nothing about improving the accuracy using polarization. This seems a poor choice of citation.

2. Pg 2 line 2: Ibrahim et al 2012 do not make any beneath surface measurements, it is entirely based on radiative transfer simulations.

3. Ibrahim et al have improved the work after 2012. A citation to the more recent work should be included:

a. A. Ibrahim, A. Gilerson, J. Chowdhary, and S. Ahmed, "Retrieval of macro- and micro-physical properties of oceanic hydrosols from polarimetric observations," Remote Sensing of Environment, vol. 186, pp. 548-566, 2016.

4. Pg2 line 6: Reduction of Sun glints: Requires more references. There is a wealth of literature on this subject beyond Zhou et al, 2017. Too many to list here.

5. Pg2 line 8-9 Insufficient citations about polarized surface reflection, see also:

a. T. Harmel et al., "Polarization impacts on the water-leaving radiance retrieval from above-water radiometric measurements," Applied Optics, vol. 51, no. 35, pp. 8324-8340, Dec 10 2012.

b. T. Harmel and M. Chami, "Estimation of the sunglint radiance field from optical satellite imagery over open ocean: Multidirectional approach and polarization aspects," Journal of Geophysical Research: Oceans, vol. 118, no. 1, pp. 76-90, 2013.

c. C. D. Mobley, "Polarized Reflectance and Transmittance Properties of Wind-blown Sea Surfaces," Applied Optics, vol. 54, no. 15, pp. 4828-4849, 2015.

d. M. Hieronymi, "Polarized reflectance and transmittance distribution functions of the ocean surface," Optics Express, vol. 24, no. 14, pp. A1045-A1068, 2016/07/11 2016.

e. R. Foster and A. Gilerson, "Polarized Transfer Functions of the Ocean Surface for Above-Surface Determination of the Vector Submarine Light Field," Applied Optics, vol. 55, no. 33, pp. 9476-9494, 11/16/2016 2016.

f. D. D'Alimonte and T. Kajiyama, "Effects of light polarization and waves slope statistics on the reflectance factor of the sea surface," Optics Express, vol. 24, no. 8, pp. 7922-
7942, 2016/04/18 2016.

6. Pg 3 line 1: There are many RT models that include light polarization:

a. B. Lafrance and M. Chami, "OSOAA (Ocean Successive Orders with Atmosphere - Advanced) Users Manual," Université Pierre et Marie Curie Laboratoire d'Océanographie de Villefranche, France 2016-11-04 2016.

b. A. A. Kokhanovsky et al., "Benchmark results in vector atmospheric radiative transfer," Journal of Quantitative Spectroscopy and Radiative Transfer, vol. 111, no. 12-13, pp. 1931-1946, 2010. [And references therein]

c. S. Korkin, A. Lyapustin, A. Sinyuk, B. Holben, and A. Kokhanovsky, "Vector radiative transfer code SORD: Performance analysis and quick start guide," Journal of Quantitative Spectroscopy and Radiative Transfer, vol. 200, pp. 295-310, 2017/10/01/ 2017.

d. Y. Ota, A. Higurashi, T. Nakajima, and T. Yokota, "Matrix formulations of radiative transfer including the polarization effect in a coupled atmosphere–ocean system," Journal of Quantitative Spectroscopy and Radiative Transfer, vol. 111, no. 6, pp. 878-894, 2010.

e. F. M. Schulz, K. Stamnes, and F. Weng, "VDISORT: An improved and generalized discrete ordinate method for polarized (vector) radiative transfer," Journal of Quantitative Spectroscopy and Radiative Transfer, vol. 61, no. 1, pp. 105-122, 1999/01/01 1999.

7. Pg 3, line 2-4. This is a false statement. Polarized radiative transfer has been happening for decades, and polarized radiative transfer of the ocean since the 1970s (Plass and Kattawar). One example:

a. G. W. Kattawar and C. N. Adams, "Stokes vector calculations of the submarine light field in an atmosphere-ocean with scattering according to a Rayleigh phase matrix: Effect of interface refractive index on radiance and polarization," Limnology and Oceanography, vol. 34, no. 8, pp. 1453-1472, 1989.

8. Pg 3, line 9: The 90 degree relative azimuth plane has been in use for a long time prior to Piskozub and Freda, see for example:

a. C. D. Mobley, "Estimation of the remote-sensing reflectance from above-surface measurements," Applied Optics, vol. 38, no. 36, pp. 7442-7455, 1999.

9. Pg 3, line 13-15. There have been many studies about the measurement and modeling of light polarization in coastal areas (to name only a few):

a. S. Sabbah, A. Barta, J. Gál, G. Horváth, and N. Shashar, "Experimental and theoretical study of skylight polarization transmitted through Snell's window of a flat water surface," Journal of the Optical Society of America A, vol. 23, no. 8, pp. 1978-1988, 2006/08/01 2006.

b. A. Tonizzo et al., "Polarized light in coastal waters: hyperspectral and multiangular analysis," Optics Express, vol. 17, no. 7, pp. 5666-5683, 2009/03/30 2009.

c. A. Tonizzo, A. Gilerson, T. Harmel, and A. Ibrahim, "Estimating particle composition and size distribution from polarized water-leaving radiance," Applied Optics, vol. 6, no. 10, pp. 5047-5058, January 2011.

d. A. Lerner, S. Sabbah, C. Erlick, and N. Shashar, "Navigation by light polarization in clear and turbid waters," Philosophical Transactions of the Royal Society B: Biological Sciences, vol. 366, no. 1565, pp. 671-679, 2011.

e. T. Harmel et al., "Polarization impacts on the water-leaving radiance retrieval from above-water radiometric measurements," Applied Optics, vol. 51, no. 35, pp. 8324-8340, Dec 10 2012.

f. Y. Gu et al., "Polarimetric imaging and retrieval of target polarization characteristics in underwater environment," Applied Optics, vol. 55, no. 3, pp. 626-637, 2016/01/20 2016.

g. A. El-habashi and S. Ahmed, "Chlorophyll fluorescence and the polarized underwater light field: comparison of vector radiative transfer simulations and multi-angular hyperspectral polarization field measurements," vol. 9827, p. 98270U, 2016.

h. J. Liu et al., "Polarization characterics of underwater, upwelling radiance of suspended particulate matters in turbid waters based on radiative transfer simulation," SPIE Remote Sensing, vol. 10784, p. 7, 2018.

i. A. C. R. Gleason, K. J. Voss, H. R. Gordon, M. S. Twardowski, and J.-F. Berthon, "Measuring and Modeling the Polarized Upwelling Radiance Distribution in Clear and Coastal Waters," Applied Sciences, vol. 8, no. 12, p. 2683, 2018.

10. Pg 4, line 18: Piskozub and Freda (2013) only write 2 paragraphs about their Monte Carlo algorithm; I would hardly call this a "description". I would have liked to see (or have been pointed to) some benchmark results comparing the code to others, so that the reader can have confidence the code is physically correct.

11. Pg 5, line 2-3: While in general the V component is negligible, the biggest source of circular polarization is total internal reflection of upwelling light by the sea surface. The off-diagonal elements in Voss and Fry are indeed zero, but this has to do with scattering only, and is not by itself justification for saying the V component is negligible.

12. Pg 5, line 6: Reflection and refraction by flat surfaces are described completely by the Fresnel equations, not 'basically'.

13. Pg 5, line 10-12: I am confused about which particle scattering Mueller matrices the authors are using. They state (line 11) that Voss and Fry 1984 is used for the water, and Volten et al (2001) is used for the aerosols. Then, the following sentence says that Mie theory is used for the phase functions. Only the (1,1) element of the Mie calculations were used? What parameters were used for the Mie calculations? How were they determined and are they representative for the Baltic Sea?

14. Pg 5, line 14: All results are specified in the principle plane. This seems to be incorrect, since polar plots of all azimuth angles are given thoughout.

15. Pg 5, line 20: There is a new 'recommended data processing' for ac-9 and ac-s instruments, the PROP-RR method, which may be applied to previously acquired data. It may be worth investigating whether this has a significant impact on the results given that the ratio of a/c is being analyzed. See:

a. N. D. Stockley, R. Röttgers, D. McKee, I. Lefering, J. M. Sullivan, and M. S. Twardowski, "Assessing uncertainties in scattering correction algorithms for reflective tube absorption measurements made with a WET Labs ac-9," Optics Express, vol. 25, no. 24, pp. A1139-A1153, 2017/11/27 2017.

16. Pg 5, line 29-31: I am very confused about these sentences. Why is aw subtracted and then added again? Perhaps subscripts should be added to clarify? (a_t) for a total, and (a_pg) particulate + CDOM absorption for a - aw. Or perhaps make it a formal equation that makes sense mathematically. Put a sigma (sum) symbol if sum is meant. Same for pg 6, line 1.

17. Pg 6, line 13: In my opinion, using the isotropic Cox-Munk slope distribution is preferable to choosing an arbitrary directional wind value. A directional wind will introduce an asymmetry into the above-surface light field, the effect of which is not being analyzed.

18. Pg 6, line 13: I would suggest choosing a more reasonable second wind speed other than 15 m/s. The limit of applicability of Cox-Munk wave slopes is 14 m/s, and when the wind is this strong, gravity waves will introduce significant uncertainty into the polarimetric measurements (in-situ) due to strong tilts in the instantaneous sea surface. TOA measurements should be unaffected, however.

19. Pg 8, Fig 2: The projection of the polar plots, or at least the zenith values of the concentric circles should be indicated.

20. Pg 8, Fig 2: The wind-speed used (5 or 15 m/s) is not indicated.

21. Pg 8, Fig 2: What aerosol optical thickness was used for the simulations at each

wavelength? What is the spectral relationship (or angstrom coefficient) used to determine it? Was it based on seasonally averaged measurements?

22. Pg 8, Fig 2c-2d: I am suspicious of the ~1000x increase ($10^3/10^0.04$) in maximum upwelling radiance from the summer to the winter. The authors should double-check that these intensities are correct.

23. Pg 8, line 1-2 : Intensity is stated to be irradiance, but units of radiance are given.

24. Pg 9, Fig 3: No wind speed, aerosol optical thickness values, or zenith labels are given.

25. Pg 13, line 11-13: I am not convinced that any increase in wind speed (and therefore an increase in the surface roughness) would cause an increase in the DoP. I just don't see any way in which a rougher surface will result in more polarization than a smooth one. Any increase in roughness should cause at least a partial de-polarization of the reflected/transmitted light field. This is also stated by the authors on pg 16, line 19. For example, see:

a. R. Foster and A. Gilerson, "Polarized Transfer Functions of the Ocean Surface for Above-Surface Determination of the Vector Submarine Light Field," Applied Optics, vol. 55, no. 33, pp. 9476-9494, 11/16/2016 2016.

26. Pg 13, line 22: "The type of water has less influence on the DoP than the season and its representative SZA." This would seem to contradict the title of the article.

27. Pg 14, line 12: This sentence seems to contradict also with the previous comment.

28. Pg 15, lines 6-11: The authors are (partially) correct that the Fresnel reflection matrix depends only the refractive index of the medium and the incidence angle (but also on the refractive index of the air, and the imaginary part of each refractive index, which governs absorption). However, the authors are incorrect to use that as justification that the observed differences come only from the water-leaving component of the radiance. The polarization of the reflected component intrinsically depends on the polarization

[I,Q,U,V] of the downwelling skylight. The reflected Stokes vector is the downwelling Stokes vector multiplied by the reflection matrix. Therefore significant DoLP variability may be introduced in the reflected component due to polarization of the skylight component, which is then combined with DoLP variability coming from the water-leaving part.

29. Pg 15, lines 16-17: Although I disagree with the reason given for the non-linearity, more importantly, the DoP may never be greater than one. If this occurs in any case, there is a significant problem with the simulations which should be addressed, or a better explanation must be given.

30. Pg 15, line 23-24: I believe the reason for the wavelength independence is because the max(DoLP) is always looking at the direct reflection of the Sun, which has little to do with the water body. See General comment #2.

31. Pg 16, line 13: Generally speaking, the DoLP tends to decrease after multiple scattering events because of the number of photons originating from different directions (and with different polarization), however the authors statement is not universally true and strongly depends on the scattering angle. For example, unpolarized light scattered by Rayleigh particles at 90 degrees becomes fully polarized. Individual scattering events often increase the polarization of the scattered light.

32. Pg 17, line 5-7: This is the expected behavior. The underwater SZA corresponding to above-water SZA of 45 and 75 degrees is ∼30 and ∼45 degrees, respectively. Since the planes of constant DoLP are orthogonal to the SZA (in single scattering), this results in a 'tilt' of the planes of constant underwater DoLP of ∼60 and ∼45 degrees (from the horizontal).

33. Pg 18, line 16-17: More likely the reason is that the measurement of Tonizzo et al, 2009 included scattering by hydrosols with different phase matrices than the Voss-Fry matrices used here.

34. Pg 19, line 7-8: The "HPR" for a flat surface, should be exactly the Brewster angle, which is ∼53 degrees, not 58. Additionally, the refractive index of the water used should be specified somewhere.

35. Pg 19, lines 5-14: I am not certain this paragraph adds anything to the discussion. I am not aware of any significance to SZA + Zenith = 2 * Brewster angle, and the "HPR" has no information about the water body, since (as defined by the authors) it is 'reflected' radiance.

36. Pg 20, line 25 to Pg 21, line 2: This statement is inaccurate. I believe there is a misunderstanding by the authors about the nature of the relationship between the reflection matrix (or Fresnel amplitude coefficients for parallel and perpendicular directions) and the reflected light field (and polarization thereof). The perpendicular and parallel Fresnel coefficients alone do not dictate the degree of polarization of reflected light. Only when they are applied to an incident light field is the DoP of the reflected light known exactly. They can say something about the possible ranges of DoP, but barring a few specific cases the actual reflected DoP may only be known after considering the coefficients and the incident light field together.

37. Pg 20, line 16-17: I disagree with this statement. When the SZA is very high (winter), the "HPM", in the authors terminology (the angles of highest underwater DoP), are allowed to propagate upward through the surface, because they fall within Snell's window (cone of angles less than the critical angle). When the Sun is higher in the sky (lower SZA), the peak DoP falls outside Snells window and is internally reflected by the sea surface, and therefore does not propagate above the water. This would seem to contradict the statement by the authors. See also Fig 4 of:

a. A. Ibrahim, A. Gilerson, T. Harmel, A. Tonizzo, J. Chowdhary, and S. Ahmed, "The relationship between upwelling underwater polarization and attenuation/absorption ratio," Optics Express, vol. 20, no. 23, pp. 25662-25680, Nov 05 2012.

38. Pg 21, line 6-9: Isn't Fig 8 a simulation of below water? This would seem to be

directly comparable with Ibrahim, 2012.

Technical Corrections

1. Pg 3 line 1: I do not see an entry in the references for Chami, 2001. Also, see specific comment #7, because this citation is out of date. (see LaFrance and Chami, 2016).
* * *

---

## Referee Comment (RC2) · Anonymous Referee #2 · 8 Mar 2019

"On the role of seawater absorption to attenuation ratio in the radiance polarization above the Southern Baltic surface" by Wlodzimierz Freda, Kamila Haule, and Slawomir Sagan

My review refers to the revised manuscript version from February 26, 2019. This version includes changes in response to the very detailed and competent first review. The suggested additional references have been included. However, at some points I would have wished more discussion with their content. Generally, the discussion comes a bit short and the first reviewer listed many reference points worth to discuss, but not mentioned in the new version (e.g. all comments >#32). I suggest adding some more

discussion and context of the findings. Specific comments: The used wind speed of 5 m/s is plausible; it's approximately the annual global mean and therefore basis of many ocean colour applications, e.g. atmospheric correction of water algorithms. In contrast, a wind speed of 15 m/s (7Bft) is typically considered as high wind, moderate or near gale, and is of less relevance for remote sensing or in situ measurements. In this case, we would have additional depolarization due to enhanced whitecap fraction (e.g. Hu et al., 2008), air bubble entrainment and possibly more sea spray generation. In the coastal regions of interest, we would not expect fully developed wind seas, but considering the large sun zenith angle of 75°, results based on the Cox-Munk model must be seen very carefully (Mobley, 2015; Hieronymi, 2016). Assuming that the applied Monte Carlo model nevertheless works properly, we will have increased multiple scattering at the sea surface in the winter case with large zenith angle. This can be an important source for depolarization. I find it not helpful to combine the effects of changing IOPs and zenith angle. The main difference in terms of season seems to be the sun zenith angle and not IOPs or ratios. There is also no need to restrict the findings to this particular region (also not in the title). Thus, it is hard to differentiate the individual effects on maximum DoP or polarization pattern.

Hu, Y., Stamnes, K., Vaughan, M., Pelon, J., Weimer, C., Wu, D., ... & Omar, A. (2008). Sea surface wind speed estimation from space-based lidar measurements. Atmospheric Chemistry and Physics, 8(13), 3593-3601.

---

## Author Response (AR1)

Dear Editors

Trying to meet the high standards of quality of the publication in the journal Ocean Science, I have twice responded to reviewers' comments. After the review made by the Reviewer 1, along with the response, I attached an updated article that included suggested corrections. Reviewer 2 reported his comments to the revised version of our article. For this reason, I enclose separate responses to two reviews by separately selecting the suggestions suggested by the Reviewer 1 and by the Reviewer 2.
* * *
Answer to Reviewer 1

First of all, we want to thank the Reviewer for their time taken to improve and comment on this publication. Below we quote the review text (in black) giving answers to each comment (in red). In the article (MS Word document) we have added comments about Specific Comments from review 1, so that you can easily identify corrected parts of the text.

Anonymous Referee #1

General Comments

The authors of the manuscript have conducted large number of polarimetric simulations of water conditions in the Baltic sea, over two seasons, and three different water types. An analysis is conducted which looks at the relationship between the absorption to attenuation of the water, and the upwelling polarization signal. It is determined that a correlation exists, and that it also depends on the Sun angle and the wind speed. A discussion about the direction of maximum DoP is also given.

The authors should be commended for the large amount of work that obviously went into this analysis, and especially for the inclusion of light polarization, which many scientists avoid.

That being said, however, I believe there to be several fundamental scientific problems with this manuscript that must be addressed before it is suitable for publication. I wish to convey to the authors that, although I have written many comments, it is because I am both familiar and passionate about the subject, and wish to see it properly conducted and thereby make an impactful contribution to the field.

Thank you for the positive and passionate feedback. Based on the specific comments the manuscript was thoroughly revised and improved. We have changed the references, we have completed the captions of the figures and the controversial fragments have simply been removed

My critique may be distilled down to a few main issues:

1. I found the literature cited by the manuscript very lacking. Almost all citations are prior to ~2012, and there has been many advances in the field in recent years.

We agree, thank you for your remark and suggestions. The citations have been revised and corrected.

2. The analysis focuses on "max(DoP)". The maximum DoP is almost always either in, or near, the specular reflection point for above water simulations due to the inherent Fresnel reflectivity of the mean sea surface. (Figs 2,3,4,5,6 of this manuscript) This is an infeasible place to be measuring polarization for ocean color, since any measured signal from the ocean will be overwhelmed by the reflectance of the Sun. Ocean color satellites will not measure at this geometry. The paper would be much more applicable if the max(DoLP) were limited to feasibly measurable angles.

It's true that our analysis focuses on "max (DoP)" and satellites scan the surface of the Earth at different zenith angles. However, as illustrated in Figures 2,3,4, we show the DoP of radiances in all directions of the hemisphere, and in Figures 5 and 6 in all directions of the principal plane. In this first approach we decided to address the commonly analyzed max(DoP) in order to find and describe seasonal correlations. In future we will extend the study to consider angles more applicable to the remote sensors.

3. I believe that the contribution made by section 3.5 of the manuscript is marginal at best. Most of the conclusions about the direction of max(DoP) are 'known', or can be determined easily from Snell's law and the knowledge that the maximum DoP occurs at scattering angles near 90 degrees. The underwater simulations are illustrative, but the explanations given for the direction of the DoP are inaccurate. See the specific comments below for further details.

We agree that the contribution of subsection 3.5 to the entire article is not very significant. For this reason, and because of the specific comments 32-38, we decided to remove this subsection.

Specific Comments

1. Pg 2 line 1-2: Garaba and Zielinkski, 2013 have very little to say about the polarization of above surface light. Nothing about improving the accuracy using polarization. This seems a poor choice of citation.

SC1. We corrected the citation.

2. Pg 2 line 2: Ibrahim et al 2012 do not make any beneath surface measurements, it is entirely based on radiative transfer simulations.

SC2. We corrected the statement.

3. Ibrahim et al have improved the work after 2012. A citation to the more recent work should be included:

a. A. Ibrahim, A. Gilerson, J. Chowdhary, and S. Ahmed, "Retrieval of macro- and micro-physical properties of oceanic hydrosols from polarimetric observations," Remote Sensing of Environment, vol.
5  186, pp. 548-566, 2016.

SC3. We included the citation accordingly.

4. Pg2 line 6: Reduction of Sun glints: Requires more references. There is a wealth of literature on this subject beyond Zhou et al, 2017. Too many to list here.

SC4. We added more recent references as suggested.

10  5. Pg2 line 8-9 Insufficient citations about polarized surface reflection, see also:

a. T. Harmel et al., "Polarization impacts on the water-leaving radiance retrieval from above-water radiometric measurements," Applied Optics, vol. 51, no. 35, pp. 8324-8340, Dec 10 2012.

b. T. Harmel and M. Chami, "Estimation of the sunglint radiance field from optical satellite imagery over open ocean: Multidirectional approach and polarization aspects,"Journal of Geophysical Research:
15  Oceans, vol. 118, no. 1, pp. 76-90, 2013.

c. C. D. Mobley, "Polarized Reflectance and Transmittance Properties of Wind-blown Sea Surfaces," Applied Optics, vol. 54, no. 15, pp. 4828-4849, 2015.

d. M. Hieronymi, "Polarized reflectance and transmittance distribution functions of the ocean surface," Optics Express, vol. 24, no. 14, pp. A1045-A1068, 2016/07/11 2016.

20  e. R. Foster and A. Gilerson, "Polarized Transfer Functions of the Ocean Surface for Above-Surface Determination of the Vector Submarine Light Field," Applied Optics, vol. 55, no. 33, pp. 9476-9494, 11/16/2016 2016.

f. D. D'Alimonte and T. Kajiyama, "Effects of light polarization and waves slope statistics on the reflectance factor of the sea surface," Optics Express, vol. 24, no. 8, pp. 7922-7942, 2016/04/18 2016.

25  SC5. We complemented the references accordingly.

6. Pg 3 line 1: There are many RT models that include light polarization:

a. B. Lafrance and M. Chami, "OSOAA (Ocean Successive Orders with Atmosphere - Advanced) Users Manual," Université Pierre et Marie Curie Laboratoire d'Océanographie de Villefranche, France 2016-11-04 2016.

b. A. A. Kokhanovsky et al., "Benchmark results in vector atmospheric radiative transfer," Journal of Quantitative Spectroscopy and Radiative Transfer, vol. 111, no. 12-13, pp. 1931-1946, 2010. [And references therein]

c. S. Korkin, A. Lyapustin, A. Sinyuk, B. Holben, and A. Kokhanovsky, "Vector radiative transfer code SORD: Performance analysis and quick start guide," Journal of Quantitative Spectroscopy and Radiative Transfer, vol. 200, pp. 295-310, 2017/10/01/ 2017.

d. Y. Ota, A. Higurashi, T. Nakajima, and T. Yokota, "Matrix formulations of radiative transfer including the polarization effect in a coupled atmosphere–ocean system," Journal of Quantitative Spectroscopy and Radiative Transfer, vol. 111, no. 6, pp. 878-894, 2010.

e. F. M. Schulz, K. Stamnes, and F. Weng, "VDISORT: An improved and generalized discrete ordinate method for polarized (vector) radiative transfer," Journal of Quantitative Spectroscopy and Radiative Transfer, vol. 61, no. 1, pp. 105-122, 1999/01/01 1999.

SC6. We corrected the statement and complemented the references accordingly.

7. Pg 3, line 2-4. This is a false statement. Polarized radiative transfer has been happening for decades, and polarized radiative transfer of the ocean since the 1970s (Plass and Kattawar). One example:

a. G. W. Kattawar and C. N. Adams, "Stokes vector calculations of the submarine light field in an atmosphere-ocean with scattering according to a Rayleigh phase matrix: Effect of interface refractive index on radiance and polarization," Limnology and Oceanography, vol. 34, no. 8, pp. 1453-1472, 1989.

SC7. We apologize for the misinformation. We corrected the statement.

8. Pg 3, line 9: The 90 degree relative azimuth plane has been in use for a long time prior to Piskozub and Freda, see for example:

a. C. D. Mobley, "Estimation of the remote-sensing reflectance from above-surface measurements," Applied Optics, vol. 38, no. 36, pp. 7442-7455, 1999.

SC*. Yes, but Mobley's Hydrolight model does not include polarization, and we show (Piskozub and Freda) that the polarization remote sensing may be useful in a plane tilted 90° from the solar azimuth angle.

9. Pg 3, line 13-15. There have been many studies about the measurement and modeling of light polarization in coastal areas (to name only a few):

a. S. Sabbah, A. Barta, J. Gál, G. Horváth, and N. Shashar, "Experimental and theoretical study of skylight polarization transmitted through Snell's window of a flat water surface," Journal of the Optical Society of America A, vol. 23, no. 8, pp. 1978-1988, 2006/08/01 2006.

b. A. Tonizzo et al., "Polarized light in coastal waters: hyperspectral and multiangular analysis," Optics Express, vol. 17, no. 7, pp. 5666-5683, 2009/03/30 2009.

c. A. Tonizzo, A. Gilerson, T. Harmel, and A. Ibrahim, "Estimating particle composition and size distribution from polarized water-leaving radiance," Applied Optics, vol. 6, no. 10, pp. 5047-5058, January 2011.

d. A. Lerner, S. Sabbah, C. Erlick, and N. Shashar, "Navigation by light polarization in clear and turbid waters," Philosophical Transactions of the Royal Society B: Biological Sciences, vol. 366, no. 1565, pp. 671-679, 2011.

e. T. Harmel et al., "Polarization impacts on the water-leaving radiance retrieval from above-water radiometric measurements," Applied Optics, vol. 51, no. 35, pp. 8324-8340, Dec 10 2012.

f. Y. Gu et al., "Polarimetric imaging and retrieval of target polarization characteristics in underwater environment," Applied Optics, vol. 55, no. 3, pp. 626-637, 2016/01/20 2016.

g. A. El-habashi and S. Ahmed, "Chlorophyll fluorescence and the polarized underwater light field: comparison of vector radiative transfer simulations and multi-angular hyperspectral polarization field measurements," vol. 9827, p. 98270U, 2016.

h. J. Liu et al., "Polarization characterics of underwater, upwelling radiance of suspended particulate matters in turbid waters based on radiative transfer simulation," SPIE Remote Sensing, vol. 10784, p. 7, 2018.

i. A. C. R. Gleason, K. J. Voss, H. R. Gordon, M. S. Twardowski, and J.-F. Berthon, "Measuring and Modeling the Polarized Upwelling Radiance Distribution in Clear and Coastal Waters," Applied Sciences, vol. 8, no. 12, p. 2683, 2018.

SC9. We corrected the paragraph and complemented the references accordingly.

10. Pg 4, line 18: Piskozub and Freda (2013) only write 2 paragraphs about their Monte Carlo algorithm; I would hardly call this a "description". I would have liked to see (or have been pointed to) some benchmark results comparing the code to others, so that the reader can have confidence the code is physically correct.

SC10. We are the users of the code, not the creators. Although we can't provide any published benchmark comparison to other such codes, we can assure about its correctness on the basis of:

- the scientific experience of the author of the code, Prof. Jacek Piskozub and the internal tests he performed in order to verify its correctness;

- the polarized radiative transfer code was written as a new version of a code used for over 20 years in works published together with world-wide ocean optics authorities (e.g. J.R, Zaneveld, D. McKee, R. Rottgers, D. Stramski)

- the unmodified version of the algorithm was successfully used, with results published in:

a. McKee D., Piskozub J., Rottgers, and R., Reynolds, R.A.: Evaluation and Improvement of an Iterative Scattering Correction Scheme for in situ Absorption and Attenuation Measurements, Journal of Atmospheric and Oceanic Technology, 30, 1527-1541, doi:10.1175/JTECH-D-12-00150.1, 2013.

b. McKee D., Piskozub J., and Brown, I.: Scattering error corrections for in situ absorption and attenuation measurements, Optics Express 16, 19480-19492, doi: 10.1364/OE.16.019480, 2008.

c. Piskozub J., and, McKee D.: Effective scattering phase functions for the multiple scattering regime, Optics Express, 19, 4786-4794, doi:10.1364/OE.19.004786, 2011.

d. Stramski D., and, Piskozub J.: Estimation of scattering error in spectrophotometric measurements of light absorption by aquatic particles from 3-D radiative transfer simulations, Applied Optics, 42, 3634-3646, doi: 10.1364/AO.42.003634, 2003.

e. Piskozub J., Flatau P.J., and Zaneveld J.R.V.: Monte Carlo study of the scattering error of a quartz reflective absorption tube, Journal of Oceanic and Atmospheric Technology, 18, 438-445, doi: 10.1175/1520-0426(2001)018<0438:MCSOTS>2.0.CO;2, 2001.

11. Pg 5, line 2-3: While in general the V component is negligible, the biggest source of circular polarization is total internal reflection of upwelling light by the sea surface. The off-diagonal elements in Voss and Fry are indeed zero, but this has to do with scattering only, and is not by itself justification for saying the V component is negligible.

SC11. We agree, corrected.

12. Pg 5, line 6: Reflection and refraction by flat surfaces are described completely by the Fresnel equations, not 'basically'.

SC12. We agree, corrected.

13. Pg 5, line 10-12: I am confused about which particle scattering Mueller matrices the authors are using. They state (line 11) that Voss and Fry 1984 is used for the water, and Volten et al (2001) is used for the aerosols. Then, the following sentence says that Mie theory is used for the phase functions. Only

the (1,1) element of the Mie calculations were used? What parameters were used for the Mie calculations? How were they determined and are they representative for the Baltic Sea?

SC13. We use Voss and Fry (1984) for seawater and Volten (2001) for aerosols. Additionally Rayleigh for molecular scattering both in atmosphere and water. The sentence about Mie theory is a mistake - has been deleted.

14. Pg 5, line 14: All results are specified in the principle plane. This seems to be incorrect, since polar plots of all azimuth angles are given thoughout.

SC14. True, sentence corrected.

15. Pg 5, line 20: There is a new 'recommended data processing' for ac-9 and ac-s instruments, the PROP-RR method, which may be applied to previously acquired data. It may be worth investigating whether this has a significant impact on the results given that the ratio of a/c is being analyzed. See:

a. N. D. Stockley, R. Röttgers, D. McKee, I. Lefering, J. M. Sullivan, and M. S. Twardowski, "Assessing uncertainties in scattering correction algorithms for reflective tube absorption measurements made with a WET Labs ac-9," Optics Express, vol. 25, no. 24, pp. A1139-A1153, 2017/11/27 2017.

SC15. Yes, we are familiar with this paper and the new correction method. However, Baltic Sea is a very unique water basin from the optical point of view and application of new method to historical data needs to be verified and validated first. As the authors conclude, " but for waters where other types of non-algal absorbing particles may be present in higher proportions, it is unclear whether the PROP-RR relationship will remain appropriate."

Our dataset includes data verified using spectroscopic methods (absorption measured with 1nm resolution - see Sagan 2008). Moreover in this study we use averaged data over seasons, therefore application of this new method (whether proper or not) to the single measurements would not affect them in a considerable degree.

16. Pg 5, line 29-31: I am very confused about these sentences. Why is aw subtracted and then added again? Perhaps subscripts should be added to clarify? (a_t) for a total, and (a_pg) particulate + CDOM absorption for a - aw. Or perhaps make it a formal equation that makes sense mathematically. Put a sigma (sum) symbol if sum is meant. Same for pg 6, line 1.

SC16. We modified the equation and the description to make it more clear, according to the suggestions. (We used the AC-9 dataset where a_w was already subtracted from the total absorption, this is why we had to add it again.)

17. Pg 6, line 13: In my opinion, using the isotropic Cox-Munk slope distribution is preferable to choosing an arbitrary directional wind value. A directional wind will introduce an asymmetry into the above-surface light field, the effect of which is not being analyzed.

SC17. The wind is directional by nature, this is why we simply decided to model the directional wind applying the same direction to all simulations. In our opinion the asymmetry of the light field caused by directional wind does not affect the above-surface light field in a significant degree.

From the previous studies (e.g. Haule et al. 2017) we know that wind direction changes can modify the remote sensing reflectance (which is proportional to water-leaving radiance) for less than 1.3% at the wind speed of 5 m/s.

The asymmetry is not clearly visible at our polar plots. That is why we do not see any need for changes.

18. Pg 6, line 13: I would suggest choosing a more reasonable second wind speed other than 15 m/s. The limit of applicability of Cox-Munk wave slopes is 14 m/s, and when the wind is this strong, gravity waves will introduce significant uncertainty into the polarimetric measurements (in-situ) due to strong tilts in the instantaneous sea surface. TOA measurements should be unaffected, however.

SC18. Cox and Munk obtained their probability density function for slopes of waves for wind speed from 1 to 14 m/s. However, several times higher values, e.g. 15 m/s, have been used in the literature:

a) Knut Stamnes, Gary E. Thomas, Jakob J. Stamnes: "Radiative Transfer in the Atmosphere and Ocean" at pages 162-163
b) Giles D'Souza, Alan S. Belward, Jean-Paul Malingreau "Advances in the Use of NOAA AVHRR Data for Land Applications" at page 82 and below.
c) Alexander Gilerson, Carlos Carrizo, Robert Foster, and Tristan Harmel, "Variability of the reflectance coefficient of skylight from the ocean surface and its implications to ocean color," Opt. Express 26, 9615-9633 (2018), doi: 10.1364/OE.26.009615
d) Doi: 10.1175/JCLI3973.1
e) Doi: 10.1175/1520-0426(2003)020<1697:MROTCD>2.0.CO;2
f) Doi: 10.1088/0256-307X/26/9/094102

Or even 20m/s:
g) Doi: 10.1088/0256-307X/26/9/094102

In our model we usually set 5 m/s as a typical (most common) wind speed in the Baltic Sea and 15 m/s as the maximal (border) value for measurements. Our results show a mathematically similar dependence between a/c ratio and max(DoP) for both wind speeds.

19. Pg 8, Fig 2: The projection of the polar plots, or at least the zenith values of the concentric circles should be indicated.

SC19. We agree, corrected.

20. Pg 8, Fig 2: The wind-speed used (5 or 15 m/s) is not indicated.

SC20. Corrected.

21. Pg 8, Fig 2: What aerosol optical thickness was used for the simulations at each wavelength? What is the spectral relationship (or angstrom coefficient) used to determine it? Was it based on seasonally averaged measurements?

SC21. We used here a single Aerosol Optical Thickness value equal to 0.12 independent of the wavelength and the same for both seasons. We did it on purpose, because introducing more realistic variability of AOT would cause an additional source of DoP changes and would not allow conclusions to be drawn about the causes of correlation.

22. Pg 8, Fig 2c-2d: I am suspicious of the 1000x increase ($10^3/10^{0.04}$) in maximum upwelling radiance from the summer to the winter. The authors should double check that these intensities are correct.

SC22. We understand the mistrust, however, the data have been checked before manuscript submission. The difference of several orders of magnitude in the maximal values of the total upwelling radiance is caused by different directions of incident sunlight (SZA). Also, the upwelling irradiance computed here contains the water-leaving part and the reflected part. The water-leaving part itself can vary of less than 1 order of magnitude due to different SZAs, but the reflected part can vary much more. For most of directions radiance intensities values are comparable.

23. Pg 8, line 1-2 : Intensity is stated to be irradiance, but units of radiance are given.

SC23. This was an error, it is radiance; corrected.

24. Pg 9, Fig 3: No wind speed, aerosol optical thickness values, or zenith labels are given.

SC24. Corrected.

25. Pg 13, line 11-13: I am not convinced that any increase in wind speed (and therefore an increase in the surface roughness) would cause an increase in the DoP. I just don't see any way in which a rougher surface will result in more polarization than a smooth one. Any increase in roughness should cause at least a partial de-polarization of the reflected/transmitted light field. This is also stated by the authors on pg 16, line 19. For example, see:

a. R. Foster and A. Gilerson, "Polarized Transfer Functions of the Ocean Surface for Above-Surface Determination of the Vector Submarine Light Field," Applied Optics, vol. 55, no. 33, pp. 9476-9494, 11/16/2016 2016.

SC25. Yes, we agree, in general the DOP decreased with wind speed. Here we speak about local and spectral values. We observed the modifications in the DOP's spectral shape at higher wind speeds but

only in backward directions relative to sun positions. This is the result of the reflection of light from higher wave slopes.

26. Pg 13, line 22: "The type of water has less influence on the DoP than the season and its representative SZA." This would seem to contradict the title of the article.

SC26. The title of the article is related to interesting correlation described in the next section. The sentence describes well the results illustrated in figure 6. In our study absorption-to-attenuation ratio varies more between seasons than between 3 considered water types within each season. We show that the character of the DoP dependence on a/c remains the same for different seasons, but the equation parameters are the same for all water types within one season.

27. Pg 14, line 12: This sentence seems to contradict also with the previous comment.

SC27. We believe that the answer above explains also this comment.

28. Pg 15, lines 6-11: The authors are (partially) correct that the Fresnel reflection matrix depends only the refractive index of the medium and the incidence angle (but also on the refractive index of the air, and the imaginary part of each refractive index, which governs absorption). However, the authors are incorrect to use that as justification that the observed differences come only from the water-leaving component of the radiance. The polarization of the reflected component intrinsically depends on the polarization [I,Q,U,V] of the downwelling skylight. The reflected Stokes vector is the downwelling Stokes vector multiplied by the reflection matrix. Therefore significant DoLP variability may be introduced in the reflected component due to polarization of the skylight component, which is then combined with DoLP variability coming from the water-leaving part.

SC28. Yes, we agree. We still believe that the impact of reflected part is minor within a season, because we set the same AOT for all simulations. However we decided to remove the paragraph as its input to the discussion is marginal.

29. Pg 15, lines 16-17: Although I disagree with the reason given for the non-linearity, more importantly, the DoP may never be greater than one. If this occurs in any case, there is a significant problem with the simulations which should be addressed, or a better explanation must be given.

SC29. We do not know the reason of nonlinearity. And we have never received the DoP value equal to one or more. We simply suspect that the physical reason for the lack of linearity can be related to the limit value of DoP, which is one.

30. Pg 15, line 23-24: I believe the reason for the wavelength independence is because the max(DoLP) is always looking at the direct reflection of the Sun, which has little to do with the water body. See General comment #2.

SC30. In my opinion the existence of the maximum DoP in this place is certainly related to the direct reflection of the Sun. However the reflection alone does not explain the existing clear correlation with IOPs.

31. Pg 16, line 13: Generally speaking, the DoLP tends to decrease after multiple scattering events because of the number of photons originating from different directions (and with different polarization), however the authors statement is not universally true and strongly depends on the scattering angle. For example, unpolarized light scattered by Rayleigh particles at 90 degrees becomes fully polarized. Individual scattering events often increase the polarization of the scattered light.

SC31. We agree that only molecular scattering (Rayleigh) raises the DoP of the beam of light. I do not consider a single photon here. In this sentence, by particle matter, we meant suspensions in sea water. However, to be exact, we change this sentence.

32. Pg 17, line 5-7: This is the expected behavior. The underwater SZA corresponding to above-water SZA of 45 and 75 degrees is 30 and 45 degrees, respectively. Since the planes of constant DoLP are orthogonal to the SZA (in single scattering), this results in a 'tilt' of the planes of constant underwater DoLP of 60 and 45 degrees (from the horizontal).

SC32. Refers to deleted subsection

33. Pg 18, line 16-17: More likely the reason is that the measurement of Tonizzo et al, 2009 included scattering by hydrosols with different phase matrices than the Voss-Fry matrices used here.

SC33. Refers to deleted subsection

34. Pg 19, line 7-8: The "HPR" for a flat surface, should be exactly the Brewster angle, which is 53 degrees, not 58. Additionally, the refractive index of the water used should be specified somewhere.

SC34. Refers to deleted subsection

35. Pg 19, lines 5-14: I am not certain this paragraph adds anything to the discussion. I am not aware of any significance to SZA + Zenith = 2 * Brewster angle, and the "HPR" has no information about the water body, since (as defined by the authors) it is 'reflected' radiance.

SC35. Refers to deleted subsection

36. Pg 20, line 25 to Pg 21, line 2: This statement is inaccurate. I believe there is a misunderstanding by the authors about the nature of the relationship between the reflection matrix (or Fresnel amplitude coefficients for parallel and perpendicular directions) and the reflected light field (and polarization thereof). The perpendicular and parallel Fresnel coefficients alone do not dictate the degree of polarization of reflected light. Only when they are applied to an incident light field is the DoP of the reflected light known exactly. They can say something about the possible ranges of DoP, but barring a

few specific cases the actual reflected DoP may only be known after consideringthe coefficients and the incident light field together.

SC36. Refers to deleted subsection

37. Pg 20, line 16-17: I disagree with this statement. When the SZA is very high (winter), the "HPM", in the authors terminology (the angles of highest underwater DoP), are allowed to propagate upward through the surface, because they fall within Snell's window (cone of angles less than the critical angle). When the Sun is higher in the sky (lower SZA), the peak DoP falls outside Snells window and is internally reflected by the sea surface, and therefore does not propagate above the water. This would seem to contradict the statement by the authors. See also Fig 4 of:

a. A. Ibrahim, A. Gilerson, T. Harmel, A. Tonizzo, J. Chowdhary, and S. Ahmed, "The relationship between upwelling underwater polarization and attenuation/absorption ratio," Optics Express, vol. 20, no. 23, pp. 25662-25680, Nov 05 2012.

SC37. Refers to deleted subsection

38. Pg 21, line 6-9: Isn't Fig 8 a simulation of below water? This would seem to be directly comparable with Ibrahim, 2012.

SC38. Refers to deleted subsection

Technical Corrections
1. Pg 3 line 1: I do not see an entry in the references for Chami, 2001. Also, see specific comment #7, because this citation is out of date. (see LaFrance and Chami, 2016)

[revised manuscript text omitted]
 polarizationabove the Southern Baltic surface" by Wlodzimierz Freda, Kamila Haule, and SlawomirSagan

5  My review refers to the revised manuscript version from February 26, 2019. This version includes changes in response to the very detailed and competent first review. The suggested additional references have been included. However, at some points I would have wished more discussion with their content. Generally, the discussion comes a bit short and the first reviewer listed many reference points worth to discuss, but not mentioned in the new version (e.g. all comments >#32). I suggest adding some moreC1discussion and

10  context of the findings. Specific comments: The used wind speed of 5m/s is plausible; it's approximately the annual global mean and therefore basis of many ocean colour applications, e.g. atmospheric correction of water algorithms. In contrast, a wind speed of 15 m/s (7Bft) is typically considered as high wind, moderate or neargale, and is of less relevance for remote sensing or in situ measurements. In this case, we would have additional depolarization due to enhanced whitecap fraction (e.g. Hu et al., 2008), air bubble entrainment

15  and possibly more sea spray generation. In the coastal regions of interest, we would not expect fully developed wind seas, but considering the large sun zenith angle of 75∘, results based on the Cox-Munk model must be seen very carefully (Mobley, 2015; Hieronymi, 2016). Assuming that the applied Monte Carlo model nevertheless works properly, we will have increased multiple scattering at the sea surface in the winter case with large zenith angle. This can be an important source for depolarization. I find it not helpful

20  to combine the effects of changing IOPs and zenith angle. The main difference in terms of season seems to be the sun zenith angle and not IOPs or ratios. There is also no need to restrict the findings to this particular region (also not in the title). Thus, it is hard to differentiate the individual effects on maximum DoP or polarization pattern.

Hu, Y., Stamnes, K., Vaughan, M., Pelon, J., Weimer, C., Wu, D., ... & Omar, A.(2008). Sea surface wind

25  speed estimation from space-based lidar measurements. Atmospheric Chemistry and Physics, 8(13), 3593-3601.

We are grateful that Reviewer 2 commented on the second version of the article in which we have included a number of corrections suggested by the Reviewer 1.

Reply to the comments contained in the review 2:

1. Assuming, according to Reviewer 1, that the contribution made by section 3.5 of the manuscript "is marginal at best and most of the conclusion are known", we have removed this part of the article without discussion with comments 32-37. We still think that the discussion of the direction of the highest polarization radiance could be meaningful for rough sea surface. However, it would require considering more SZAs, and that is why we decided to include such discussion in a separate article.

2. The comment regarding additional depolarization at low sun position and strong wind (high waves) is helpful. Our Monte Carlo algorithm does not take into account wave heights, and whitecaps, but only the slope of the wave during transmission/reflection from the surface. We do not take into account the additional depolarization effect and information about this has been added to the article (see red text added at pages: 5, 17 and 21)

3. We agree that the results of our modeling should be universal, independent of the region. However, our IOPs came from measurements made in the Southern Baltic, where the absorption and attenuation values are high in comparison to typical oceanic waters, that is why we decided to keep this information in the title.

4. Separating the impact of IOPs and the angular position of the sun is a good idea. However, we originally wanted to show the seasonal variability of the upwelling polarization radiance above the surface of the Baltic Sea. In the shortest days, the sun reaches, unfortunately, only 12 degrees above the horizon there. We therefore decided there is no sense to use the variability of the sun position for winter IOPs.

5. An additional discussion with the content of the cited articles suggested by reviewer 1 has been added. (See red text at pages 2 and 3)

The text of the article after corrections has been added below.

[revised manuscript text omitted]